# Exploring the impact of child underweight status on common childhood illnesses among children under five years in Bangladesh along with spatial analysis

**Khondokar Naymul Islam**[1], **Sumaya Sultana**[2], **Ferdous Rahman**[3], **Abdur Rahman**[1]*

1 Statistics Discipline, Science, Engineering and Technology School, Khulna University, Khulna, Bangladesh, 2 Faculty of Science, Department of Statistics, Bangabandhu Sheikh Mujibur Rahman Science & Technology University, Pirojpur, Bangladesh, 3 Faculty of Social Science, Department of Public Administration, Bangabandhu Sheikh Mujibur Rahman Science & Technology University, Gopalganj, Dhaka, Bangladesh

* rahman@stat.ku.ac.bd

**Data Availability Statement:** The data underlying the results presented in the study are available from the DHS authority from: https://dhsprogram.com.

## Abstract

### Background

In developing countries like Bangladesh, under-five children are mostly experiencing and suffering from common diseases like fever, cough, diarrhea, and acute respiratory infections (ARI). To mitigate these problems, it's crucial to spot prevalent areas and take proper action. This study investigates the spatial distribution and associated factors of prevalent childhood illnesses across Bangladesh.

### Methods and findings

This research comprised 8,306 children's information from the Bangladesh Demographic and Health Survey (BDHS) 2017–18. We performed chi-square, t-tests, binary logistic regression and spatial analyses in this work. BDHS survey data and GPS data were aggregated to identify common childhood illnesses among under-five children. Moran's index first mapped childhood illnesses. Afterward, Getis-Ord Gi* discovered hot and cold spots for illnesses. However, Kriging interpolation predicted child illnesses in unsampled areas. Here, 33.2% (CI: 32.2–34.3), 36% (CI: 35–37.1), 4.7% (CI: 4.3–5.2), and 12.9% (CI: 12.2–13.6) of children under five had fever, cough, diarrhea, and ARI, respectively. In the fortnight before to the survey, 47.3% (CI: 46.2–48.3) of under-5 children were ill. Common childhood illnesses are associated with children's (age, underweight status, etc.), mothers' (age, education, etc.), and household factors (residency, wealth index, etc.). Underweight status is associated with fever, cough and at least one disease. The unsampled north-western and south-western areas of Bangladesh had a higher prevalence of fever, cough, ARI and at least one common disease. Cough was most common in the central-northern region; fever was most common in the lower southern region; and ARI was most common in

**Funding:** The author(s) received no specific funding for this work.

**Competing interests:** The authors have declared that no competing interests exist.

Bangladesh's south-east. Childhood diseases were more prevalent in Bangladesh's central-northern and southern regions.

## Conclusions

Our research demonstrates the regional clustering of common childhood diseases in Bangladesh. Policymakers should focus on these higher-prevalence regions, and the necessary preventive measures should be taken immediately.

## Introduction

Childhood diseases are an unavoidable part of growing up, and many children face a variety of health issues throughout their formative years. According to the World Health Organization (WHO), around 2.6 million under-five children died, the majority of which were caused by preventable or treatable causes [1]. Childhood illnesses are a significant worry in densely populated, underdeveloped nations such as Bangladesh [2–4]. Acute respiratory infections (ARIs), fever, cough, and diarrhea are among the most prevalent diseases in children [5,6]. For both parents and their children, these circumstances may result in pain and worry [7–9].

Among children's common diseases, ARI accounts for 15% of mortality in children under the age of five globally, with the highest incidence seen in low- and middle-income countries (LMICs) [10]. In low- and middle-income countries (LMICs), atmospheric air pollution levels exceed the WHO recommendations for about 98% of children under the age of five [11]. In high-income nations, 52% of children fall into this category [11]. Bangladesh is classified as a low- and middle-income country (LMIC), with over 166 million people (63% of the total population) living in rural regions [12]. Although other infectious illnesses have decreased, the BDHS-2014 reports that ARIs remain a major cause of death in Bangladesh, responsible for 21% of all deaths [13]. The BDHS 2017–18 data shows a 25% mortality rate due to ARI, a rise from the 2014 data [14]. On the other hand, over 40% of children have access to adequate healthcare for treating diarrhea, while approximately 60% have access to good treatment for ARI symptoms on a global scale. Notwithstanding these statistics, diarrhea and ARI remain the primary factors contributing to mortality in children under the age of 5 [15,16]. R. A. Anne et al. (2018) conducted a study that revealed that over 20% of children under five in Bangladesh suffered from cold and fever, and an additional 16.69% experienced diarrhea [17]. This highlights the notable frequency of these health problems among young children in the country. In addition, childhood fever, which is the most prevalent clinical presentation among children under the age of five, serves as a notable indication of the public health consequences linked to these diseases [18].

Several studies have shown that inadequate nutrition in childhood, namely being underweight, may have adverse impacts on a child's physical and mental development [13,19]. This can increase their vulnerability to communicable diseases and other serious infections, resulting in a higher risk of mortality. Consequently, these adverse effects impose a higher economic burden on societies. Furthermore, children with inadequate nutritional status are more vulnerable to severe manifestations of common infectious diseases, and these prevalent illnesses often lead to nutritional insufficiency-caused fatalities. Malnutrition is responsible for almost 45% of deaths in children between the ages of 0 and 59 months in Bangladesh, making it a major leading cause of mortality [9,19–22]. It is widely recognized that malnutrition in

children and mothers has a detrimental effect on growth and development in the fields of national and international economics, as well as health and development [22].

Historically, multiple researchers have sought to discover the underlying risk factors for childhood diseases using various methodologies and statistical models [3,19,23,24]. In essence, it can be said that frequent childhood diseases pose significant barriers to the attainment of the third objective of the Sustainable Development Goals (SDG), which centers on the advancement of optimal health and well-being [25,26]. To successfully attain the Sustainable Development Goals, Bangladesh must prioritize the mitigation of prevalent diseases among children and the implementation of disease monitoring. This involves the identification of locations with a high risk, evaluation of vaccination coverage and deficiencies, development of strategies for interventions, and allocation of resources appropriately. Presenting this information graphically through spatial distribution would greatly enhance its accessibility and convenience.

The evidence that we have at our disposal indicates a lack of research on the influence of underweight children on common childhood diseases in Bangladesh, particularly in terms of spatial distribution. Our objective in this study is to assess the association of child underweight with the occurrence of major childhood illnesses, both individually and for at least one of the four disorders (fever, cough, diarrhea, and ARI), among under-5 children in Bangladesh. We aim to address the lack of crucial knowledge in this area. Another primary goal is to use spatial techniques to improve our understanding of the geographical distribution of common childhood diseases and their hotspot regions. This scientific research seeks to reveal new perspectives, provide empirical data, and contribute to the existing knowledge base in the domain of child health and nutrition. Our study's expected results will provide valuable insights for developing policies, interventions, and strategies aimed at enhancing child health and reducing the negative effects of child malnutrition on prevalent childhood diseases in Bangladesh.

## Methods and materials

### Sampling

We have used a secondary data set obtained from a country representative from the Bangladesh Demographic and Health Survey (BDHS) 2017–18. The survey was intended to assess a number of health metrics and provide a comprehensive portrait of Bangladesh's population, which focuses on issues related to maternal and child health in particular. The sampling frame for the BDHS was constructed based on the collection of enumeration areas (EAs) from the 2011 census of people and housing in the Democratic People's Republic of Bangladesh, and the EA was the primary sampling unit for this investigation. The survey employed a two-stage stratified sampling technique. In the initial phase, a total of 675 EAs were selected, with 227 and 448 from urban and rural areas, respectively. Nevertheless, a natural disaster rendered it impossible to gather data from three EAs. For the second round of sampling, a systematic sample of 30 homes was chosen from each EA. Consequently, a response rate of 99% was achieved by interviewing 20,127 of the chosen 20,376 ever-married women aged 15–49. Furthermore, to avoid any bias in the selection process, we conducted interviews with all women between the ages of 15 and 49 who were ever married and had children under the age of five without replacing any participants from the preselected homes. The data collection of children's demographic, health, and nutritional information was conducted with the verbal consent of women. To ensure accurate national representation, sample weights were calculated for each sampling stage and cluster. These weights were adjusted due to uneven allocation in divisions, urban and rural areas, and variations in response rates. The adjustments addressed issues such as underestimating variability and correcting for under- and over-sampling within specific

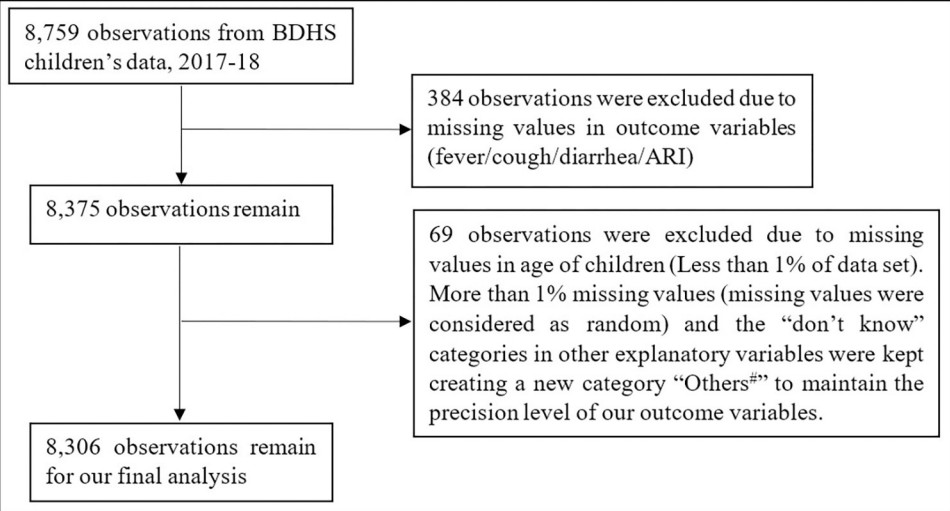

**Fig 1. Study population and sample selection.**

groups. The BDHS report provides a comprehensive explanation of this weighting process. The 2017–2018 BDHS also collected data on the geographical locations of the EAs' centers. While doing the spatial analysis, these locations on the map were conceived of. The genuine locations of the questioned clusters in metropolitan areas were randomly altered by no more than two kilometers (km) in order to protect the respondents', their own, and their communities' whereabouts. This was done to ensure the respondents' privacy. Clusters found in rural regions were randomly moved up to 5 kilometers away, and an additional 1% of clusters chosen at random from rural areas were moved up to 10 kilometers away. A thorough explanation of the survey, including details on the procedure for gathering data and determining the sample size, was provided in the BDHS 2017–18 report [27].

## Inclusion criteria

This study was open to females who had ever been married, were between the ages of 15 and 49, and had already given birth to at least one child. Data from a total of 8,759 mother-child couples was examined in the present research. After removing the missing values from our study variables and those that made up less than 1% of the data set for any explanatory variable, the total number of observations was 8,306. To ensure that the prevalence of our study variables was appropriately reflected for all of the explanatory factors, missing values that were more than 1% of the data set, as well as the categories for "don't know", were merged into a new category named "Others#" (Fig 1).

## Variables

**Outcome variables.** We examined five outcome variables: fever, cough, diarrhea, acute respiratory infections (ARI), and at least one common childhood disease (i.e., fever, cough, diarrhea, and ARI) in the fortnight before the survey among under-five children. The BDHS assessed the children's health condition based on the mother's response to the survey question, "Has your child experienced diarrhea, cough, or fever in the past two weeks?" The mothers also responded to the question about their children's respiratory concerns to detect the presence of ARI. Therefore, the BDHS dataset included self-reported confirmations of their

children's diseases. Those who suffer from at least one of the four common diseases mentioned earlier will be identified as either exposed to a disease coded "1" or unexposed to any disease coded "0."

**Explanatory variables.**   Based on prior research, we considered certain factors related to the characteristics of children, their mothers, and their households. Children's characteristics include age (in months), sex, underweight, BCG vaccination (received at any time before the survey), and birth order. On the other hand, the characteristics of mothers include factors such as mother's age (in years), maternal educational level, early age at first birth, maternal working status, exposure to mass media, and religion. Additionally, household characteristics include factors such as household size, wealth index, place of residence, and drinking water source [23,28–32]. The child's and mother's ages are continuous in type, while the other explanatory variables are categorical.

This particular study focused solely on the anthropometric index weight-for-age to assess the underweight status of children under the age of five in Bangladesh. We classified the children as underweight if their weight for age (WAZ) was less than -2 SD (standard deviation), and not underweight otherwise [33]. Here, the birth order is categorized as 1, 2–4, 5 or more. The age of 19 or earlier for conceiving the first child for a mother was considered the early age at first birth [34]. The variable mass media exposure was coded as 'yes' if a respondent was exposed to at least one of the three media: reading a newspaper or magazine, listening to the radio, and watching television (barely once a week); otherwise, 'no'. Household size is classified into three categories: '1–4', '5–8', and 'more than 8'. When it comes to drinking water sources, the category "any other" encompasses the use of water from various sources such as pipes into the dwelling, pipes to the yard, plot, or neighbor, public taps, standpipes, protected or unprotected wells, rivers, dams, lakes, ponds, streams, canals, rainwater, tanker trucks, carts with small tanks, or bottled water. Child's age, sex of child, BCG vaccination, mother's age, maternal educational level, working status, wealth index and place of residence remain as they are in the BDHS dataset.

## Statistical analysis

Descriptive statistics of each of the selected variables and the distribution of common childhood diseases by different factors were shown with a 95% confidence interval (calculated using STATA software) by adjusting the sampling weight. We conducted a binary logistic regression model to evaluate the relationship between selected explanatory variables and dichotomous outcome variables. Data pertaining to prevalent illnesses among the subjects were interconnected with Global Positioning System (GPS) data, providing geographic information about their respective localities. The spatial data for each study cluster were generated using ArcGIS 10.8 software. To ascertain the spatial distribution of childhood illnesses, a spatial autocorrelation model called Moran's index was employed to quantify patterns and trends in common childhood illnesses. A positive value of Moran's index indicates a tendency toward clustering. Conversely, a negative Moran's index suggests a tendency for dispersion, indicating a spatial pattern where similar values are scattered apart [35].

To determine the spatial heterogeneity of significantly high or low occurrences of common childhood illnesses within each cluster, the Getis-Ord Gi* statistic tool in ArcGIS was used. Furthermore, we used the Kriging interpolation technique, which has a low mean square error and residual [36], to forecast the prevalence of fever, cough, diarrhea, ARI, and exposure to at least one of these four diseases in regions where the study samples were not obtained. The interpolation process generated smooth maps by predicting the proportion of outcome variables in unsampled locations, specifically within enumeration areas. This interpolation was

performed using a general formula for calculating a weighted sum of the available data:

$$\hat{Z}(S_0) = \sum_{i=1}^{N} \lambda_i Z(S_i) \tag{1}$$

Where:

$Z(S_i)$ = the measured value at the i[th] location

$\lambda_i$ = an unknown weight for the measured value at the ith location

$(S_0)$ = the prediction location

$N$ = the number of measured values

We cleaned up the dataset and did some preliminary statistical analyses using STATA 17 (MP). These included frequency distributions of the variables we were interested in, bivariate associations between dependent and independent variables, and odds ratios adjusting the confounding variables at a 5% significance level. Microsoft Office Excel 21 and ArcGIS 10.8 software are used to perform the geo-spatial analyses.

## Results

### Univariate analysis

The weighted and unweighted frequencies of the selected variables were displayed in univariate analysis for each of the categories of the outcome and explanatory variables. In this research, the weighted mean age of the children was 28.70 (±17.45) months, while the mean age of the children's mother was 25.71(±5.66) years among the weighted sample of 8312 children. This study had a slightly higher proportion of male children (52.16%) than its counterpart. About 26.33% of the respondents lived in cities, with the remaining individuals residing in rural areas. Apparently, one-fifth of the children (21.06%) were found to be underweight. More than two-thirds (79.67%) of respondents picked tube wells as their primary source of drinking water (Table 1). In children under the age of five, 47.25% were diagnosed with at least one of the four diseases: fever, ARI, diarrhea and cough. We found that 36.03%, 33.23%, 12.86%, and 4.74% of children experienced cough, fever, ARI, and diarrhea, respectively. While only a small number of children experienced diarrhea, the other illnesses were widespread among them.

### Bivariate analysis

In bivariate analysis, the chi-square test was used to determine the association between each dependent and categorical explanatory variable, and for continuous explanatory variables, a t-test was used. We added all significant variables (p-value 0.05) to the multivariate analysis and discarded insignificant variables from further analysis. The bivariate analysis findings are presented in Table 2. The results depict the percentage estimates of 8,312 (weighted) study children with their illness status for fever, cough, diarrhea, ARI and disease (Table 2). As shown in Table 3, the weighted interaction percentages with a 95% CI of the categorical explanatory variables and the weighted mean of continuous explanatory variables with a standard deviation for each study variable are calculated.

### Findings from multivariate analysis

We used the multivariable binary logistic regression model with the BDHS survey data to estimate the effects of maternal and children's characteristics on childhood diseases. Table 4 shows from an inspection of this study that children from Muslim households were 1.31 (CI: 1.08–1.6) times more likely to suffer from fever compared to children from non-Muslim families. Children who remained in the underweight category were 1.34 times more likely to

**Table 1. Baseline characteristics of the study participants.**

| Variables | Categories | Unweighted (8,306) | Weighted (8,312) |
|---|---|---|---|
| | | N (%)/Mean (±SD) | N (%)/Mean (±SD) |
| **Outcome variables** | | | |
| Fever | No | 5,560 (66.94) | 5,550 (66.77) |
| | Yes | 2,746 (33.06) | 2,762 (33.23) |
| Cough | No | 5,281 (63.58) | 5,317 (63.97) |
| | Yes | 3,025 (36.42) | 2,995 (36.03) |
| Diarrhea | No | 7,896 (95.06) | 7,918 (95.26) |
| | Yes | 410 (4.94) | 394 (4.74) |
| ARI | No | 7,259 (87.39) | 7,243 (87.14) |
| | Yes | 1,047 (12.61) | 1,069 (12.86) |
| Disease | No | 4,378 (52.71) | 4,384 (52.75) |
| | Yes | 3,928 (47.29) | 3,928 (47.25) |
| **Explanatory variables** | | | |
| Child's age (in months) | Continuous | 28.79 (17.57) | 28.70 (17.45) |
| Sex of child | Male | 4,326 (52.08) | 4,335 (52.16) |
| | Female | 3,980 (47.92) | 3,977 (47.84) |
| Religion | Non-Muslim | 700 (8.43) | 669 (8.05) |
| | Muslim | 7,606 (91.57) | 7,643 (91.95) |
| Underweight child | No | 6,242 (75.15) | 6,284 (75.60) |
| | Yes | 1,804 (21.72) | 1,750 (21.06) |
| | Others# | 260 (3.13) | 278 (3.34) |
| Received BCG vaccination | No | 367 (4.42) | 353 (4.25) |
| | Yes | 4,715 (56.77) | 4,758 (57.25) |
| | Others# | 3,224 (38.82) | 3,201 (38.50) |
| Birth order | 1 | 3,183 (38.32) | 3,187 (38.34) |
| | 2–4 | 4,683 (56.38) | 4,695 (56.49) |
| | 5+ | 440 (5.30) | 430 (5.18) |
| Mother's age (in years) | Continuous | 25.81 (5.68) | 25.71 (5.66) |
| Maternal educational level | No education | 591 (7.12) | 593 (7.13) |
| | Primary | 2,383 (28.69) | 2,359 (28.38) |
| | Secondary | 3,914 (47.12) | 4,045 (48.66) |
| | Higher | 1,418 (17.07) | 1,315 (15.83) |
| Early age at first birth | No | 2,509 (30.21) | 2,394 (28.81) |
| | Yes | 5,797 (69.79) | 5,918 (71.19) |
| Working status | No | 4,957 (59.68) | 4,967 (59.76) |
| | Yes | 3,349 (40.32) | 3,345 (40.24) |
| Mass media exposure | No | 2,989 (35.99) | 2,892 (34.79) |
| | Yes | 5,317 (64.01) | 5,420 (65.21) |
| Household size | 1–4 | 2,630 (31.66) | 2,733 (32.88) |
| | 5–8 | 4,421 (53.23) | 4,360 (52.46) |
| | More than 8 | 1,255 (15.11) | 1,219 (14.66) |
| Wealth index | Poorest | 1,824 (21.96) | 1,781 (21.43) |
| | Poorer | 1,666 (20.06) | 1,690 (20.33) |
| | Middle | 1,482 (17.84) | 1,569 (18.88) |
| | Richer | 1,636 (19.70) | 1,653 (19.88) |
| | Richest | 1,698 (20.44) | 1,619 (19.47) |

(*Continued*)

**Table 1.** (Continued)

| Variables | Categories | Unweighted (8,306) | Weighted (8,312) |
|---|---|---|---|
| | | N (%)/Mean (±SD) | N (%)/Mean (±SD) |
| Place of residence | Urban | 2,868 (34.53) | 2,239 (26.93) |
| | Rural | 5,438 (65.47) | 6,073 (73.07) |
| Drinking water source | Tube well | 6,624 (79.75) | 6,622 (79.67) |
| | Any other | 724 (8.72) | 710 (8.55) |
| | Others# | 958 (11.53) | 980 (11.79) |

#Others included missing values and don't know.

experience fever than children who were not underweight (AOR = 1.34; CI = 1.18–1.51). Shockingly, children who received BCG vaccination were more likely to have a fever, with 2.7 times higher odds than children who did not receive BCG vaccination (CI: 1.98–3.68). The children whose mothers completed their secondary education were found to have a 26% (AOR = 1.26; CI: 1.01–1.58) higher likelihood of getting fever compared to the children whose mothers had no education.

In the logistic regression model with the outcome variable cough, it was found that male children had an 18% higher chance of suffering from cough than female children (AOR = 1.18; CI: 1.07–1.31). This chance of suffering from cough was 15% (AOR = 1.15; CI: 1.02–1.30) higher among underweight children compared to not-underweight children. Similar to suffering from fever, children who received BCG vaccination were found to have a higher likelihood (AOR = 2.86; CI: 2.09–3.91) of suffering from cough than children who did not receive BCG vaccination. The sampled children with mothers with secondary education had 1.39 (AOR = 1.39; CI: 1.12–1.27) times higher odds of getting coughs than the children whose mothers had no education. Children from households with one to four members had a 15% (AOR = 1.15; CI: 1.02–1.29) higher likelihood of suffering from cough than those from households with five to eight members. Children from those families who were drinking water from a tube well were 41% (AOR = 1.41; CI: 1.16–1.7) more likely to cough than the children who were drinking water from any other water source.

Sampled children from Muslim families were 78% (AOR = 1.78; CI: 1.09–2.89) more likely to have diarrhea than children from non-Muslim families. Children who had the BCG vaccine had 7.24 times (AOR = 7.24; CI: 2.95–17.76) greater likelihood of being exposed to diarrhea than children who did not get the BCG immunization. There was a correlation between diarrhea and drinking water. More specifically, children from households drinking from tube wells were 68% (AOR = 1.68; CI: 1.04–2.7) more likely to have diarrhea.

Tuning other diseases here, we also get that male children were likely to suffer 1.41 (AOR = 1.41; CI: 1.22–1.64) times more from ARI than their counterparts. Children who received BCG vaccination showed a 3.24 (AOR = 3.24; CI: 1.99–5.27) times higher likelihood of being exposed to ARI than children who were not vaccinated. Children whose mothers were under 20 years old at the time of their first birth had a 38% (AOR = 1.38; CI: 1.15–1.66) higher risk of developing ARI than children whose mothers were at or over 20 years old at the time of their first delivery. We also found that the chance of getting affected by ARI is 28% (AOR = 1.28; CI: 1.1–1.49) higher for children whose mothers were working than for children whose mothers were not working. Children from families that picked a tube well as their drinking water source were 1.50 (AOR = 1.50; CI: 1.09–2.07) times more likely to be impacted by ARI than children from families who chose any other water source for drinking water.

**Table 2. Bivariate association of children's, their mother's and household characteristics with fever, cough, ARI, diarrhea and at least one disease among under 5 children in Bangladesh, BDHS-2017-18.**

| Factors | Having Fever | | p-value | Having Cough | | p-value | Having Diarrhea | | p-value | Having ARI | | p-value | Having Disease | | p-value |
|---|---|---|---|---|---|---|---|---|---|---|---|---|---|---|---|
| | No | Yes | | No | Yes | | No | Yes | | No | Yes | | No | Yes | |
| N (%) | 5560 (66.94) | 2746 (33.06) | | 5281 (63.58) | 3025 (36.42) | | 7896 (95.06) | 410 (4.94) | | 7259 (87.39) | 1047 (12.61) | | 4378 (52.71) | 3928 (47.29) | |
| **Child's age in months, mean (±SD)** | | | <0.001* | | | <0.001* | | | <0.001* | | | <0.001* | | | <0.001* |
| | 29.8 (17.8) | 26.7 (16.8) | | 29.4 (17.9) | 27.6 (17.0) | | 29.2 (17.7) | 21.7 (13.7) | | 29.4 (17.6) | 24.3 (16.5) | | 30.2 (18.0) | 27.2 (17.0) | |
| **Sex of child** | | | 0.078 | | | **0.002*** | | | **0.038*** | | | **<0.001*** | | | **0.002*** |
| Male | 2858 (66.07) | 1468 (33.93) | | 2681 (61.97) | 1645 (38.03) | | 4092 (94.59) | 234 (5.41) | | 3711 (85.78) | 615 (14.22) | | 2210 (51.09) | 2116 (48.91) | |
| Female | 2702 (67.89) | 1278 (32.11) | | 2600 (65.33) | 1380 (34.67) | | 3804 (95.58) | 176 (4.42) | | 3548 (89.15) | 432 (10.85) | | 2168 (54.47) | 1812 (45.53) | |
| **Religion** | | | **0.003*** | | | 0.415 | | | **0.022*** | | | 0.272 | | | **0.009*** |
| Non-Muslim | 504 (72) | 196 (28) | | 455 (65) | 245 (35) | | 678 (96.86) | 22 (3.14) | | 621 (88.71) | 79 (11.29) | | 402 (57.43) | 298 (42.57) | |
| Muslim | 5056 (66.47) | 2550 (33.53) | | 4826 (63.45) | 2780 (36.55) | | 7218 (94.9) | 388 (5.1) | | 6638 (87.27) | 968 (12.73) | | 3976 (52.27) | 3630 (47.73) | |
| **Underweight child** | | | **<0.001*** | | | **<0.001*** | | | 0.71 | | | **0.002*** | | | **<0.001*** |
| No | 4246 (68.02) | 1996 (31.98) | | 3987 (63.87) | 2255 (36.13) | | 5931 (95.02) | 311 (4.98) | | 5451 (87.33) | 791 (12.67) | | 3326 (53.28) | 2916 (46.72) | |
| Yes | 1109 (61.47) | 695 (38.53) | | 1102 (61.09) | 702 (38.91) | | 1715 (95.07) | 89 (4.93) | | 1563 (86.64) | 241 (13.36) | | 879 (48.73) | 925 (51.27) | |
| Others# | 205 (78.85) | 55 (21.15) | | 192 (73.85) | 68 (26.15) | | 250 (96.15) | 10 (3.85) | | 245 (94.23) | 15 (5.77) | | 173 (66.54) | 87 (33.46) | |
| **Received BCG vaccination** | | | **<0.001*** | | | **<0.001*** | | | **<0.001*** | | | **<0.001*** | | | **<0.001*** |
| No | 288 (78.47) | 79 (21.53) | | 293 (79.84) | 74 (20.16) | | 361 (98.37) | 6 (1.63) | | 340 (92.64) | 27 (7.36) | | 264 (71.93) | 103 (28.07) | |
| Yes | 2943 (62.42) | 1772 (37.58) | | 2808 (59.55) | 1907 (40.45) | | 4379 (92.87) | 336 (7.13) | | 3979 (84.39) | 736 (15.61) | | 2210 (46.87) | 2505 (53.13) | |
| Others# | 2329 (72.24) | 895 (27.76) | | 2180 (67.62) | 1044 (32.38) | | 3156 (97.89) | 68 (2.11) | | 2940 (91.19) | 284 (8.81) | | 1904 (59.06) | 1320 (40.94) | |
| **Birth order** | | | **0.017*** | | | 0.34 | | | 0.19 | | | 0.28 | | | 0.24 |
| 1 | 2190 (68.8) | 993 (31.2) | | 2023 (63.56) | 1160 (36.44) | | 3028 (95.13) | 155 (4.87) | | 2804 (88.09) | 379 (11.91) | | 1689 (53.06) | 1494 (46.94) | |
| 2–4 | 3083 (65.83) | 1600 (34.17) | | 2964 (63.29) | 1719 (36.71) | | 4442 (94.85) | 241 (5.15) | | 4076 (87.04) | 607 (12.96) | | 2442 (52.15) | 2241 (47.85) | |
| 5+ | 287 (65.23) | 153 (34.77) | | 294 (66.82) | 146 (33.18) | | 426 (96.82) | 14 (3.18) | | 379 (86.14) | 61 (13.86) | | 247 (56.14) | 193 (43.86) | |
| **Mother's age in years, mean (±SD)** | | | 0.11 | | | **0.001*** | | | **<0.001*** | | | **<0.001*** | | | **<0.001*** |
| | 25.9 (5.7) | 25.7 (5.6) | | 26.0 (5.7) | 25.5 (5.6) | | 25.9 (5.7) | 24.7 (5.2) | | 25.9 (5.7) | 25.2 (5.5) | | 26.1 (5.8) | 25.5 (5.6) | |
| **Maternal educational level** | | | **<0.001*** | | | **<0.001*** | | | 0.83 | | | **0.029*** | | | **<0.001*** |
| No education | 402 (68.02) | 189 (31.98) | | 403 (68.19) | 188 (31.81) | | 558 (94.42) | 33 (5.58) | | 520 (87.99) | 71 (12.01) | | 331 (56.01) | 260 (43.99) | |
| Primary | 1565 (65.67) | 818 (34.33) | | 1518 (63.7) | 865 (36.3) | | 2262 (94.92) | 121 (5.08) | | 2083 (87.41) | 300 (12.59) | | 1276 (53.55) | 1107 (46.45) | |
| Secondary | 2577 (65.84) | 1337 (34.16) | | 2415 (61.7) | 1499 (38.3) | | 3725 (95.17) | 189 (4.83) | | 3386 (86.51) | 528 (13.49) | | 1969 (50.31) | 1945 (49.69) | |
| Higher | 1016 (71.65) | 402 (28.35) | | 945 (66.64) | 473 (33.36) | | 1351 (95.28) | 67 (4.72) | | 1270 (89.56) | 148 (10.44) | | 802 (56.56) | 616 (43.44) | |
| **Early age at first birth** | | | **0.003*** | | | **0.005*** | | | 0.08 | | | **<0.001*** | | | **0.003*** |

*(Continued)*

**Table 2.** (Continued)

| Factors | Having Fever | | p-value | Having Cough | | p-value | Having Diarrhea | | p-value | Having ARI | | p-value | Having Disease | | p-value |
|---|---|---|---|---|---|---|---|---|---|---|---|---|---|---|---|
| | No | Yes | | No | Yes | | No | Yes | | No | Yes | | No | Yes | |
| No | 1739 (69.31) | 770 (30.69) | | 1652 (65.84) | 857 (34.16) | | 2401 (95.7) | 108 (4.3) | | 2259 (90.04) | 250 (9.96) | | 1384 (55.16) | 1125 (44.84) | |
| Yes | 3821 (65.91) | 1976 (34.09) | | 3629 (62.6) | 2168 (37.4) | | 5495 (94.79) | 302 (5.21) | | 5000 (86.25) | 797 (13.75) | | 2994 (51.65) | 2803 (48.35) | |
| **Working status** | | | 0.17 | | | **0.035*** | | | 0.11 | | | **0.006*** | | | 0.92 |
| No | 3289 (66.35) | 1668 (33.65) | | 3197 (64.49) | 1760 (35.51) | | 4697 (94.75) | 260 (5.25) | | 4373 (88.22) | 584 (11.78) | | 2615 (52.75) | 2342 (47.25) | |
| Yes | 2271 (67.81) | 1078 (32.19) | | 2084 (62.23) | 1265 (37.77) | | 3199 (95.52) | 150 (4.48) | | 2886 (86.17) | 463 (13.83) | | 1763 (52.64) | 1586 (47.36) | |
| **Mass media exposure** | | | 0.18 | | | 0.91 | | | 0.064 | | | **0.008*** | | | 0.76 |
| No | 1973 (66.01) | 1016 (33.99) | | 1898 (63.5) | 1091 (36.5) | | 2859 (95.65) | 130 (4.35) | | 2574 (86.12) | 415 (13.88) | | 1582 (52.93) | 1407 (47.07) | |
| Yes | 3587 (67.46) | 1730 (32.54) | | 3383 (63.63) | 1934 (36.37) | | 5037 (94.73) | 280 (5.27) | | 4685 (88.11) | 632 (11.89) | | 2796 (52.59) | 2521 (47.41) | |
| **Household size** | | | 0.68 | | | **0.011*** | | | 0.97 | | | 0.62 | | | **0.047*** |
| 1–4 | 1743 (66.27) | 887 (33.73) | | 1612 (61.29) | 1018 (38.71) | | 2498 (94.98) | 132 (5.02) | | 2291 (87.11) | 339 (12.89) | | 1335 (50.76) | 1295 (49.24) | |
| 5–8 | 2973 (67.25) | 1448 (32.75) | | 2850 (64.47) | 1571 (35.53) | | 4205 (95.11) | 216 (4.89) | | 3861 (87.33) | 560 (12.67) | | 2362 (53.43) | 2059 (46.57) | |
| More than 8 | 844 (67.25) | 411 (32.75) | | 819 (65.26) | 436 (34.74) | | 1193 (95.06) | 62 (4.94) | | 1107 (88.21) | 148 (11.79) | | 681 (54.26) | 574 (45.74) | |
| **Wealth index** | | | **<0.001*** | | | 0.37 | | | 0.077 | | | **<0.001*** | | | **0.034*** |
| Poorest | 1188 (65.13) | 636 (34.87) | | 1136 (62.28) | 688 (37.72) | | 1732 (94.96) | 92 (5.04) | | 1556 (85.31) | 268 (14.69) | | 942 (51.64) | 882 (48.36) | |
| Poorer | 1103 (66.21) | 563 (33.79) | | 1051 (63.09) | 615 (36.91) | | 1586 (95.2) | 80 (4.8) | | 1434 (86.07) | 232 (13.93) | | 865 (51.92) | 801 (48.08) | |
| Middle | 980 (66.13) | 502 (33.87) | | 949 (64.04) | 533 (35.96) | | 1393 (93.99) | 89 (6.01) | | 1319 (89) | 163 (11) | | 767 (51.75) | 715 (48.25) | |
| Richer | 1066 (65.16) | 570 (34.84) | | 1034 (63.2) | 602 (36.8) | | 1574 (96.21) | 62 (3.79) | | 1420 (86.8) | 216 (13.2) | | 850 (51.96) | 786 (48.04) | |
| Richest | 1223 (72.03) | 475 (27.97) | | 1111 (65.43) | 587 (34.57) | | 1611 (94.88) | 87 (5.12) | | 1530 (90.11) | 168 (9.89) | | 954 (56.18) | 744 (43.82) | |
| **Place of residence** | | | **<0.001*** | | | 0.65 | | | 0.8 | | | **0.002*** | | | **0.041*** |
| Urban | 1994 (69.53) | 874 (30.47) | | 1833 (63.91) | 1035 (36.09) | | 2724 (94.98) | 144 (5.02) | | 2552 (88.98) | 316 (11.02) | | 1556 (54.25) | 1312 (45.75) | |
| Rural | 3566 (65.58) | 1872 (34.42) | | 3448 (63.41) | 1990 (36.59) | | 5172 (95.11) | 266 (4.89) | | 4707 (86.56) | 731 (13.44) | | 2822 (51.89) | 2616 (48.11) | |
| **Drinking water source** | | | **<0.001*** | | | **<0.001*** | | | **0.006*** | | | **<0.001*** | | | **<0.001*** |
| Tube well | 4376 (66.06) | 2248 (33.94) | | 4145 (62.58) | 2479 (37.42) | | 6272 (94.69) | 352 (5.31) | | 5748 (86.78) | 876 (13.22) | | 3407 (51.43) | 3217 (48.57) | |
| Any other | 529 (73.07) | 195 (26.93) | | 513 (70.86) | 211 (29.14) | | 701 (96.82) | 23 (3.18) | | 669 (92.4) | 55 (7.6) | | 443 (61.19) | 281 (38.81) | |
| Others# | 655 (68.37) | 303 (31.63) | | 623 (65.03) | 335 (34.97) | | 923 (96.35) | 35 (3.65) | | 842 (87.89) | 116 (12.11) | | 528 (55.11) | 430 (44.89) | |

#Others included missing values and don't know

*p-value represents significant variable at a 5% significance level; CI = Confidence Interval.

Male children showed a 16% (AOR = 1.16; CI: 1.05–1.28) higher chance of getting ill from at least one common childhood illness than their counterparts. The likelihood of illness in Muslim children was 24% (AOR = 1.24; CI: 1.04–1.49) higher than in non-Muslim children.

**Table 3. Weighted prevalence of fever, cough, diarrhea, ARI and at least one common disease among under 5 children based on children's and their mother's characteristics with 95% CI.**

| Factors | Having Fever | Having Cough | Having Diarrhea | Having ARI | Disease |
|---|---|---|---|---|---|
| **% (95% CI)** | 33.2 (32.2–34.3) | 36 (35–37.1) | 4.7 (4.3–5.2) | 12.9 (12.2–13.6) | 47.3 (46.2–48.3) |
| **Child's age in months, mean (±SD)** | | | | | |
| | 26.72 (16.71) | 27.5 (16.84) | 21.54 (13.22) | 24.12 (16.11) | 27.07 (16.85) |
| **Sex of child** | | | | | |
| Male | 54.0 (52.1–55.9) | 54.8 (53–56.6) | 56.1 (51.1–60.9) | 56.1 (51.1–60.9) | 54.1 (52.5–55.6) |
| Female | 46.0 (44.1–47.9) | 45.2 (43.4–47) | 43.9 (39.1–48.9) | 43.9 (39.1–48.9) | 45.9 (44.4–47.5) |
| **Religion** | | | | | |
| Non-Muslim | 6.7 (5.8–7.7) | 7.8 (6.9–8.8) | 4.8 (3.1–7.4) | 4.8 (3.1–7.4) | 7.2 (6.4–8) |
| Muslim | 93.3 (92.3–94.2) | 92.2 (91.2–93.1) | 95.2 (92.6–96.9) | 95.2 (92.6–96.9) | 92.8 (92–93.6) |
| **Underweight child** | | | | | |
| No | 73.7 (72.1–75.3) | 75.4 (73.8–76.9) | 75.1 (70.6–79.2) | 75.1 (70.6–79.2) | 75 (73.7–76.4) |
| Yes | 24.3 (22.7–25.9) | 22.4 (21–24) | 22.1 (18.2–26.5) | 22.1 (18.2–26.5) | 22.8 (21.5–24.1) |
| Others[#] | 2.0 (1.5–2.6) | 2.2 (1.7–2.8) | 2.8 (1.5–5) | 2.8 (1.5–5) | 2.2 (1.8–2.7) |
| **Received BCG vaccination** | | | | | |
| No | 2.6 (2.1–3.3) | 2.3 (1.8–2.9) | 1 (0.4–2.7) | 1 (0.4–2.7) | 2.4 (2–3) |
| Yes | 64.8 (63.0–66.6) | 63.7 (62–65.4) | 83.2 (79.2–86.6) | 83.2 (79.2–86.6) | 64.5 (63–66) |
| Others[#] | 32.5 (30.8–34.3) | 34 (32.3–35.7) | 15.8 (12.5–19.8) | 15.8 (12.5–19.8) | 33.1 (31.6–34.6) |
| **Birth order** | | | | | |
| 1 | 36.5 (34.7–38.3) | 38.9 (37.1–40.6) | 37.9 (33.2–42.8) | 37.9 (33.2–42.8) | 38.5 (37–40) |
| 2–4 | 57.9 (56.0–59.7) | 56.3 (54.6–58.1) | 58.9 (53.9–63.6) | 58.9 (53.9–63.6) | 56.5 (55–58.1) |
| 5+ | 5.6 (4.8–6.6) | 4.8 (4.1–5.6) | 3.3 (1.9–5.6) | 3.3 (1.9–5.6) | 5 (4.3–5.7) |
| **Mother's age in years, mean (±SD)** | | | | | |
| | 25.58 (5.62) | 25.38 (5.61) | 24.52 (5.07) | 25.08 (5.53) | 25.39 (5.54) |
| **Maternal educational level** | | | | | |
| No education | 6.6 (5.7–7.6) | 6 (5.2–6.9) | 9.5 (7–12.8) | 9.5 (7–12.8) | 6.5 (5.8–7.3) |
| Primary | 29.0 (27.3–30.7) | 28 (26.4–29.7) | 27.7 (23.5–32.4) | 27.7 (23.5–32.4) | 27.6 (26.2–29) |
| Secondary | 50.7 (48.9–52.6) | 51.6 (49.8–53.4) | 47.5 (42.6–52.4) | 47.5 (42.6–52.4) | 51.4 (49.8–52.9) |
| Higher | 13.7 (12.4–15.0) | 14.4 (13.2–15.7) | 15.3 (12.1–19.2) | 15.3 (12.1–19.2) | 14.5 (13.4–15.6) |
| **Early age at first birth** | | | | | |
| No | 26.8 (25.2–28.5) | 26.8 (25.2–28.4) | 24.4 (20.4–28.9) | 24.4 (20.4–28.9) | 27.1 (25.8–28.5) |
| Yes | 73.2 (71.5–74.8) | 73.2 (71.6–74.8) | 75.6 (71.1–79.6) | 75.6 (71.1–79.6) | 72.9 (71.5–74.2) |
| **Working status** | | | | | |
| No | 60.9 (59.0–62.7) | 58.6 (56.8–60.3) | 63.3 (58.4–67.9) | 63.3 (58.4–67.9) | 59.9 (58.3–61.4) |
| Yes | 39.1 (37.3–41.0) | 41.4 (39.7–43.2) | 36.7 (32.1–41.6) | 36.7 (32.1–41.6) | 40.1 (38.6–41.7) |
| **Mass media exposure** | | | | | |
| No | 35.4 (33.7–37.2) | 34.8 (33.1–36.5) | 31.8 (27.3–36.5) | 31.8 (27.3–36.5) | 34.4 (32.9–35.9) |
| Yes | 64.6 (62.8–66.3) | 65.2 (63.5–66.9) | 68.2 (63.5–72.7) | 68.2 (63.5–72.7) | 65.6 (64.1–67.1) |
| **Household size** | | | | | |
| 1–4 | 33.7 (31.9–35.4) | 34.6 (32.9–36.3) | 32.6 (28.1–37.4) | 32.6 (28.1–37.4) | 34 (32.6–35.5) |
| 5–8 | 52.2 (50.3–54.1) | 51.4 (49.6–53.1) | 53.4 (48.5–58.3) | 53.4 (48.5–58.3) | 52 (50.4–53.6) |
| More than 8 | 14.1 (12.9–15.5) | 14.1 (12.9–15.4) | 14 (10.9–17.8) | 14 (10.9–17.8) | 14 (12.9–15.1) |
| **Wealth index** | | | | | |
| Poorest | 22.2 (20.6–23.7) | 22.3 (20.8–23.8) | 22.2 (18.4–26.6) | 22.2 (18.4–26.6) | 21.8 (20.6–23.1) |
| Poorer | 20.5 (19.0–22.0) | 20.7 (19.3–22.2) | 20.1 (16.5–24.4) | 20.1 (16.5–24.4) | 20.5 (19.3–21.8) |
| Middle | 19.7 (18.3–21.3) | 19 (17.6–20.4) | 23.6 (19.6–28) | 23.6 (19.6–28) | 19.6 (18.4–20.9) |
| Richer | 21.1 (19.6–22.6) | 19.8 (18.4–21.3) | 14.1 (11–17.9) | 14.1 (11–17.9) | 20.1 (18.9–21.4) |

*(Continued)*

**Table 3.** (Continued)

| Factors | Having Fever | Having Cough | Having Diarrhea | Having ARI | Disease |
|---|---|---|---|---|---|
| Richest | 16.6 (15.2–18.0) | 18.3 (17–19.7) | 20 (16.3–24.2) | 20 (16.3–24.2) | 17.9 (16.7–19.1) |
| **Place of residence** | | | | | |
| Urban | 25.2 (23.6–26.8) | 26.3 (24.8–28) | 25.8 (21.7–30.3) | 25.8 (21.7–30.3) | 26 (24.6–27.3) |
| Rural | 74.8 (73.2–76.4) | 73.7 (72–75.2) | 74.2 (69.7–78.3) | 74.2 (69.7–78.3) | 74 (72.7–75.4) |
| **Drinking water source** | | | | | |
| Tube well | 81.5 (80.0–82.9) | 81.7 (80.3–83.1) | 85 (81.1–88.2) | 85 (81.1–88.2) | 81.5 (80.3–82.7) |
| Any other | 7.1 (6.2–8.1) | 6.7 (5.8–7.6) | 5.5 (3.6–8.2) | 5.5 (3.6–8.2) | 7 (6.3–7.9) |
| Others# | 11.4 (10.3–12.7) | 11.6 (10.5–12.8) | 9.5 (7–12.8) | 9.5 (7–12.8) | 11.4 (10.5–12.5) |

#Others included missing values and don't know; CI = Confidence Interval.

Children who were underweight had 1.24 (AOR = 1.24; CI: 1.10–1.4) times higher likelihood of contracting common illnesses than children who were not underweight. Remarkably, the results indicated that children who had received the BCG vaccine had 3.43 (AOR = 3.43; CI: 2.58–4.55) times higher chances of having any common childhood illnesses compared to those who had not gotten the immunization. The mother's secondary school education was found to have a negative impact on common childhood illnesses. Children whose mothers completed secondary school had a 28% (AOR = 1.28; CI: 1.04–1.58) higher likelihood of being sick than children whose mothers had no education at all. The source of drinking water also influences the likelihood that a child will contract at least one common childhood ailment. Children from households whose major drinking water source is a tube well had a 37% (AOR = 1.37; CI: 1.14–1.65) greater likelihood of contracting sickness than children from other families.

## The geographical distribution of outcome variables (fever, cough, diarrhea, ARI and at least one childhood common childhood disease)

Fig 2 shows the distribution of the prevalence of fever, cough, diarrhea, ARI and common childhood illnesses. Here, the gray, blue, yellow, green, and red dots represent the prevalence rates of fever, cough, diarrhea, ARI, and common childhood illnesses at 0%, 1%-25%, 26%-50%, 51%-75%, and 76%-100%, respectively, on maps with corresponding names.

## Spatial pattern identification for fever, cough, diarrhea, ARI and at least one childhood common childhood disease among under five children in Bangladesh

Spatial analyses, like hotspot analysis and spatial interpolation, need a clustered pattern of data with geographical coordinates instead of a random pattern. In our study, we used Global Moran's I to measure the spatial autocorrelation to find this pattern. The Moran's index value for fever in children younger than five was 0.18, with a Z score of 6.87 (Fig 3A). This is significant at the 1% (P-value<0.001) level, which means that we can be 99% sure that the frequency of fever is clustered with geographic coordinates (since the Moran's index is positive). At the 1% significance level, cough (Moran's index 0.11; z-score 4.33), ARI (Moran's index 0.10; z-score 3.76) and at least one common childhood disease (Moran's index 0.16; z-score 6.04) were also not spread out randomly (Fig 3B, 3D and 3E). For diarrhea, the sick children were randomly distributed (Moran's index = 0.04, Z-score = 1.56, P = 0.19) (Fig 3C) and therefore further

**Table 4. Binary logistic regression result for each of the five outcome variables.**

| Factors | Fever | | Cough | | Diarrhea | | ARI | | Common disease | |
|---|---|---|---|---|---|---|---|---|---|---|
| | COR (95% CI) | AOR (95% CI) | COR (95% CI) | AOR (95% CI) | COR (95% CI) | AOR (95% CI) | COR (95% CI) | AOR (95% CI) | COR (95% CI) | AOR (95% CI) |
| **Child's age in months (Continuous)** | | | | | | | | | | |
| | 0.99 (0.99–0.99) | 0.99 (0.99–1.00) | 0.99 (0.99–1.00) | 1 (0.99–1.00) | 0.97 (0.97–0.98) | 0.99 (0.98–1.00) | 0.98 (0.98–0.99) | 0.98 (0.98–0.99) | 0.99 (0.99–0.99) | 0.99 (0.99–1.00) |
| **Sex of child** | | | | | | | | | | |
| Female | - | - | ref. | ref. | ref. | ref. | ref. | ref. | ref. | ref. |
| Male | - | - | 1.18 (1.07–1.30) | 1.18 (1.07–1.31) | 1.18 (0.94–1.48) | 1.16 (0.92–1.46) | 1.40 (1.21–1.61) | 1.41 (1.22–1.64) | 1.16 (1.05–1.27) | 1.16 (1.05–1.28) |
| **Religion** | | | | | | | | | | |
| Non-Muslim | ref. | ref. | | - | ref. | ref. | - | - | ref. | ref. |
| Muslim | 1.34 (1.10–1.62) | 1.31 (1.08–1.60) | - | - | 1.78 (1.1–2.89) | 1.78 (1.09–2.89) | - | - | 1.25 (1.05–1.48) | 1.24 (1.04–1.49) |
| **Underweight child** | | | | | | | | | | |
| No | ref. | ref. | ref. | ref. | - | - | ref. | ref. | ref. | ref. |
| Yes | 1.29 (1.15–1.46) | 1.34 (1.18–1.51) | 1.11 (0.99–1.25) | 1.15 (1.02–1.30) | - | - | 1.02 (0.86–1.21) | 1.08 (0.90–1.29) | 1.18 (1.05–1.33) | 1.24 (1.10–1.40) |
| Others[#] | 0.52 (0.37–0.73) | 0.61 (0.43–0.86) | 0.56 (0.41–0.77) | 0.64 (0.46–0.88) | - | - | 0.40 (0.22–0.73) | 0.49 (0.27–0.90) | 0.50 (0.37–0.68) | 0.59 (0.44–0.80) |
| **Received BCG vaccination** | | | | | | | | | | |
| No | ref. | ref. | ref. | ref. | ref. | ref. | ref. | ref. | ref. | ref. |
| Yes | 2.31 (1.73–3.09) | 2.70 (1.98–3.68) | 2.76 (2.04–3.71) | 2.86 (2.09–3.91) | 6.5 (2.67–15.78) | 7.24 (2.95–17.76) | 2.64 (1.67–4.18) | 3.24 (1.99–5.27) | 3.07 (2.35–4.00) | 3.43 (2.58–4.55) |
| Others[#] | 1.49 (1.11–2.01) | 2.26 (1.51–3.37) | 1.92 (1.42–2.60) | 2.21 (1.49–3.28) | 1.74 (0.69–4.37) | 2.72 (0.98–7.57) | 1.30 (0.81–2.08) | 2.62 (1.42–4.82) | 1.84 (1.41–2.41) | 2.59 (1.79–3.74) |
| **Birth order** | | | | | | | | | | |
| 1 | ref. | ref. | - | - | - | - | | - | - | - |
| 2–4 | 1.12 (1.00–1.24) | 1.09 (0.98–1.22) | - | - | - | - | | - | - | - |
| 5+ | 1.23 (0.97–1.55) | 1.20 (0.94–1.55) | - | - | - | - | | - | - | - |
| **Mother's age in years (Continuous)** | | | | | | | | | | |
| | - | - | 0.98 (0.98–0.99) | 1.00 (0.99–1.01) | 0.96 (0.94–0.98) | 0.98 (0.96–0.99) | 0.98 (0.96–0.99) | 1.00 (0.98–1.01) | 0.98 (0.97–0.99) | 0.99 (0.99–1.00) |
| **Maternal educational level** | | | | | | | | | | |
| No education | ref. | ref. | ref. | ref. | - | - | ref. | ref. | ref. | ref. |
| Primary | 1.15 (0.93–1.43) | 1.20 (0.96–1.49) | 1.28 (1.03–1.59) | 1.25 (1.00–1.55) | - | - | 1.03 (0.76–1.39) | 1.01 (0.74–1.38) | 1.12 (0.92–1.37) | 1.10 (0.89–1.35) |
| Secondary | 1.19 (0.97–1.46) | 1.26 (1.01–1.58) | 1.44 (1.17–1.77) | 1.39 (1.12–1.72) | - | - | 1.16 (0.87–1.55) | 1.20 (0.88–1.64) | 1.31 (1.08–1.59) | 1.28 (1.04–1.58) |
| Higher | 0.90 (0.72–1.14) | 1.08 (0.82–1.41) | 1.13 (0.90–1.43) | 1.18 (0.92–1.50) | - | - | 0.94 (0.67–1.30) | 1.12 (0.77–1.63) | 1.01 (0.81–1.25) | 1.08 (0.84–1.38) |
| **Early age at first birth** | | | | | | | | | | |
| No | ref. | ref. | ref. | ref. | - | - | ref. | ref. | ref. | ref. |
| Yes | 1.16 (1.04–1.30) | 1.06 (0.94–1.20) | 1.17 (1.05–1.31) | 1.12 (0.99–1.27) | - | - | 1.42 (1.20–1.68) | 1.38 (1.15–1.66) | 1.17 (1.05–1.30) | 1.10 (0.97–1.24) |
| **Working status** | | | | | | | | | | |
| No | - | - | ref. | ref. | - | - | ref. | ref. | - | - |
| Yes | - | - | 1.08 (0.98–1.19) | 1.08 (0.98–1.20) | - | - | 1.25 (1.08–1.44) | 1.28 (1.10–1.49) | - | - |

*(Continued)*

**Table 4.** (Continued)

| Factors | Fever | | Cough | | Diarrhea | | ARI | | Common disease | |
|---|---|---|---|---|---|---|---|---|---|---|
| | COR (95% CI) | AOR (95% CI) | COR (95% CI) | AOR (95% CI) | COR (95% CI) | AOR (95% CI) | COR (95% CI) | AOR (95% CI) | COR (95% CI) | AOR (95% CI) |
| **Mass media exposure** | | | | | | | | | | |
| No | - | - | - | - | - | - | ref. | ref. | - | - |
| Yes | - | - | - | - | - | - | 0.86 (0.74–0.99) | 0.93 (0.78–1.11) | - | - |
| **Household size** | | | | | | | | | | |
| 5–8 | - | - | ref. | ref. | - | - | - | - | ref. | ref. |
| 1–4 | - | - | 1.12 (1.00–1.25) | 1.15 (1.02–1.29) | - | - | - | - | 1.09 (0.98–1.21) | 1.12(1.00–1.25) |
| More than 8 | - | - | 0.97 (0.84–1.12) | 1.00 (0.85–1.16) | - | - | - | - | 0.93 (0.81–1.07) | 0.96 (0.83–1.11) |
| **Wealth index** | | | | | | | | | | |
| Poorest | ref. | ref. | - | - | - | - | ref. | ref. | ref. | ref. |
| Poorer | 0.96 (0.82–1.12) | 0.94 (0.80–1.09) | - | - | - | - | 0.92 (0.75–1.13) | 0.90 (0.73–1.12) | 0.98 (0.85–1.14) | 0.93 (0.80–1.09) |
| Middle | 1.02 (0.87–1.19) | 1.02 (0.87–1.21) | - | - | - | - | 0.70 (0.56–0.88) | 0.71 (0.56–0.92) | 1.04 (0.90–1.21) | 1.00 (0.85–1.17) |
| Richer | 1.04 (0.89–1.21) | 1.07 (0.90–1.26) | - | - | - | - | 0.87 (0.71–1.08) | 0.96 (0.75–1.22) | 0.99 (0.85–1.15) | 0.97 (0.82–1.14) |
| Richest | 0.75 (0.64–0.88) | 0.85 (0.70–1.04) | - | - | - | - | 0.68 (0.54–0.85) | 0.87 (0.65–1.16) | 0.83 (0.71–0.96) | 0.89 (0.74–1.07) |
| **Place of residence** | | | | | | | | | | |
| Urban | ref. | ref. | - | - | - | - | ref. | ref. | ref. | ref. |
| Rural | 1.14 (1.02–1.28) | 1.02 (0.90–1.16) | - | - | - | - | 1.23 (1.05–1.45) | 1.02 (0.85–1.23) | 1.10 (0.99–1.22) | 0.97 (0.86–1.09) |
| **Drinking water source** | | | | | | | | | | |
| Any Other | ref. | ref. | ref. | ref. | ref. | ref. | ref. | ref. | | ref. |
| Tube well | 1.35 (1.12–1.63) | 1.18 (0.96–1.44) | 1.49 (1.24–1.80) | 1.41 (1.16–1.70) | 1.7 (1.06–2.72) | 1.68 (1.04–2.70) | 1.69 (1.24–2.29) | 1.50 (1.09–2.07) | 1.47 (1.24–1.75) | 1.37 (1.14–1.65) |
| Others[#] | 1.25 (0.98–1.58) | 1.12 (0.87–1.44) | 1.39 (1.11–1.75) | 1.41 (1.11–1.80) | 1.27 (0.7–2.29) | 1.11 (0.61–2.02) | 1.60 (1.11–2.30) | 1.46 (0.99–2.13) | 1.33 (1.07–1.65) | 1.29 (1.02–1.64) |

#Others included missing values and don't know; CI = Confidence Interval; COR = Crude Odds ratio; AOR = Adjusted Odds ratio.

spatial analyses (hot-spot analysis and Kriging interpolation technique) were not performed for diarrhea.

## Hotspot analysis of child malnutrition on common childhood illnesses among children under five years in Bangladesh

We conducted a hot-spot analysis using spatial autocorrelation to identify fever, cough, ARI, and at least one common childhood disease among children under five. Hot spots and cold spots were generated based on available observation samples from specific geographical locations. The hot spot areas for fever and cough were mainly identified in the north-western and central-southern parts, while the cold spots were identified in the central, south-western and south-eastern parts. The hot spot areas for cough were found in the north-western and central southern parts while the cold spots were identified in the central, south-western and south-eastern parts.

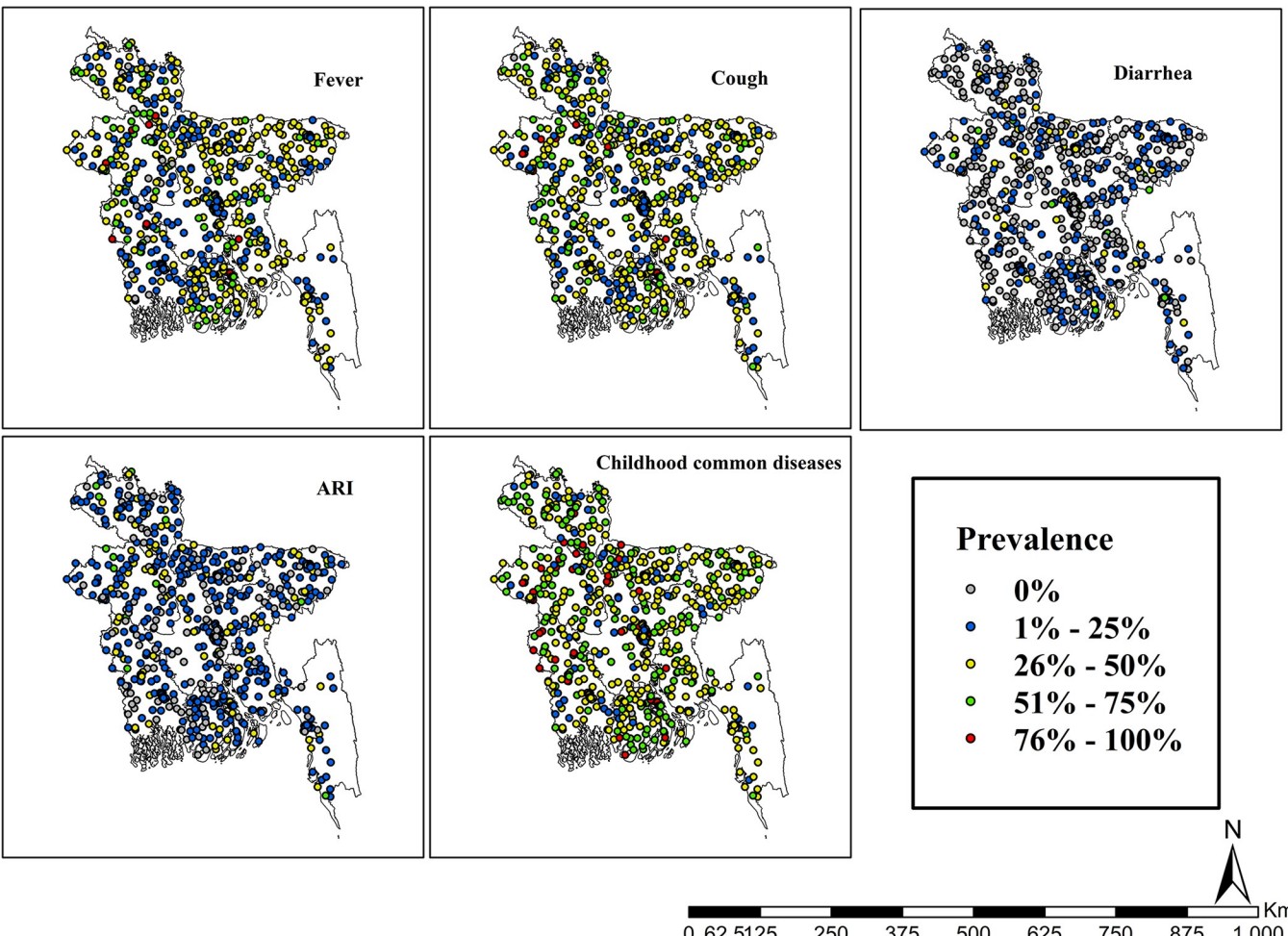

**Fig 2. Geographic distribution of fever, cough, diarrhea, ARI and childhood common diseases in Bangladesh.** The shape files are freely accessed using the following link: https://data.humdata.org/dataset/cod-ab-bgd.

The hot spot areas for ARI were found in the north-western part, and the cold spots were in the central to south-western part of the country. Therefore, the remaining cold spots are situated in the central to south-western part of the observed country. The hot spot areas for at least one common childhood illness were found in the north-western and south-eastern parts of the country, and the cold spots were spotted in the central to south-western and south-eastern parts of the country (Fig 4). Most of these hotspots for all the considered diseases were in the Rangpur, Rajshahi, and Barisal divisions. While some were present in Khulna, Chattogram, and Mymensingh, some rare hotspots, especially for fever and ARI, were found in the Dhaka and Sylhet divisions.

## Interpolation of prevalence of common childhood illnesses

In contrast to the hotspot analysis using the Kriging interpolation technique, we estimated the prevalence of diseases in unsampled areas using data from sampled areas. This led to a higher prevalence of fever in the areas without sampled data in the north-western, south-western, and lower southern parts of Bangladesh, while the south-western and slightly south-eastern parts showed a lower prevalence. The majority of the estimated areas with a higher prevalence of

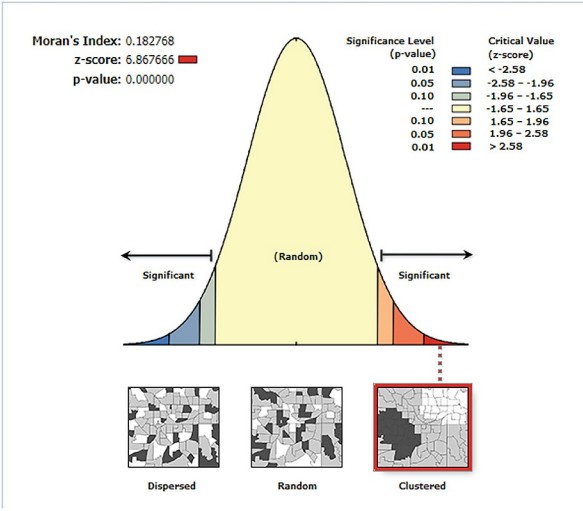

(a) : Spatial pattern of fever

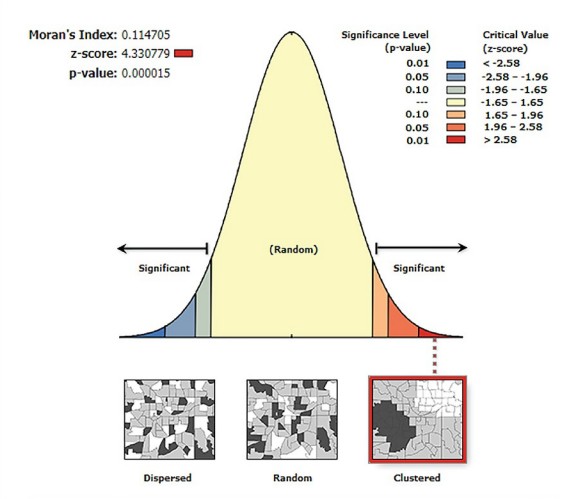
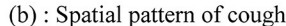

(b) : Spatial pattern of cough

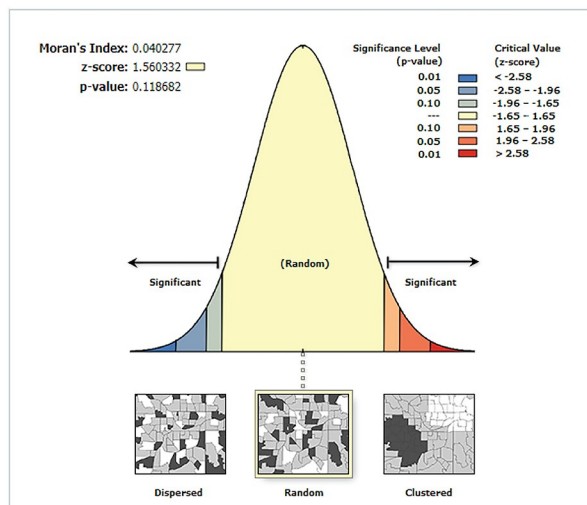

(c) : Spatial pattern of diarrhea

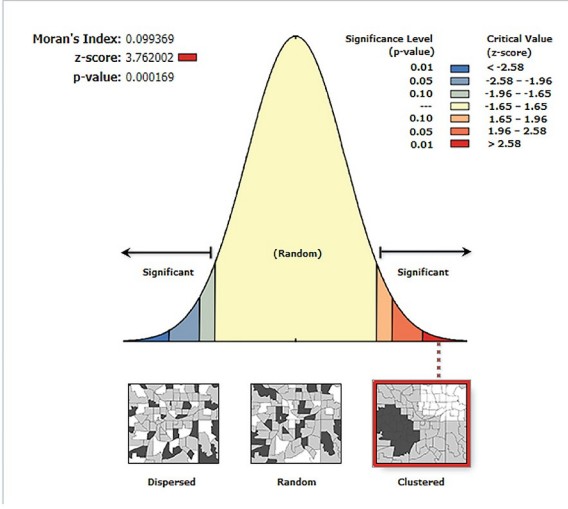

(d) : Spatial pattern of ARI

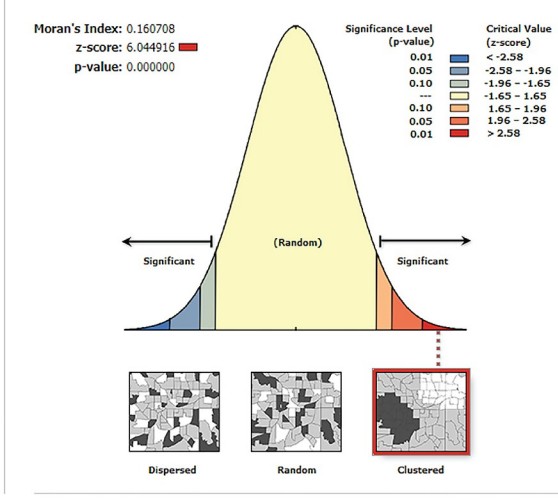

(e) : Spatial pattern of childhood common disease

**Fig 3. Spatial pattern of fever, cough, diarrhea, ARI and at least one childhood common disease.**

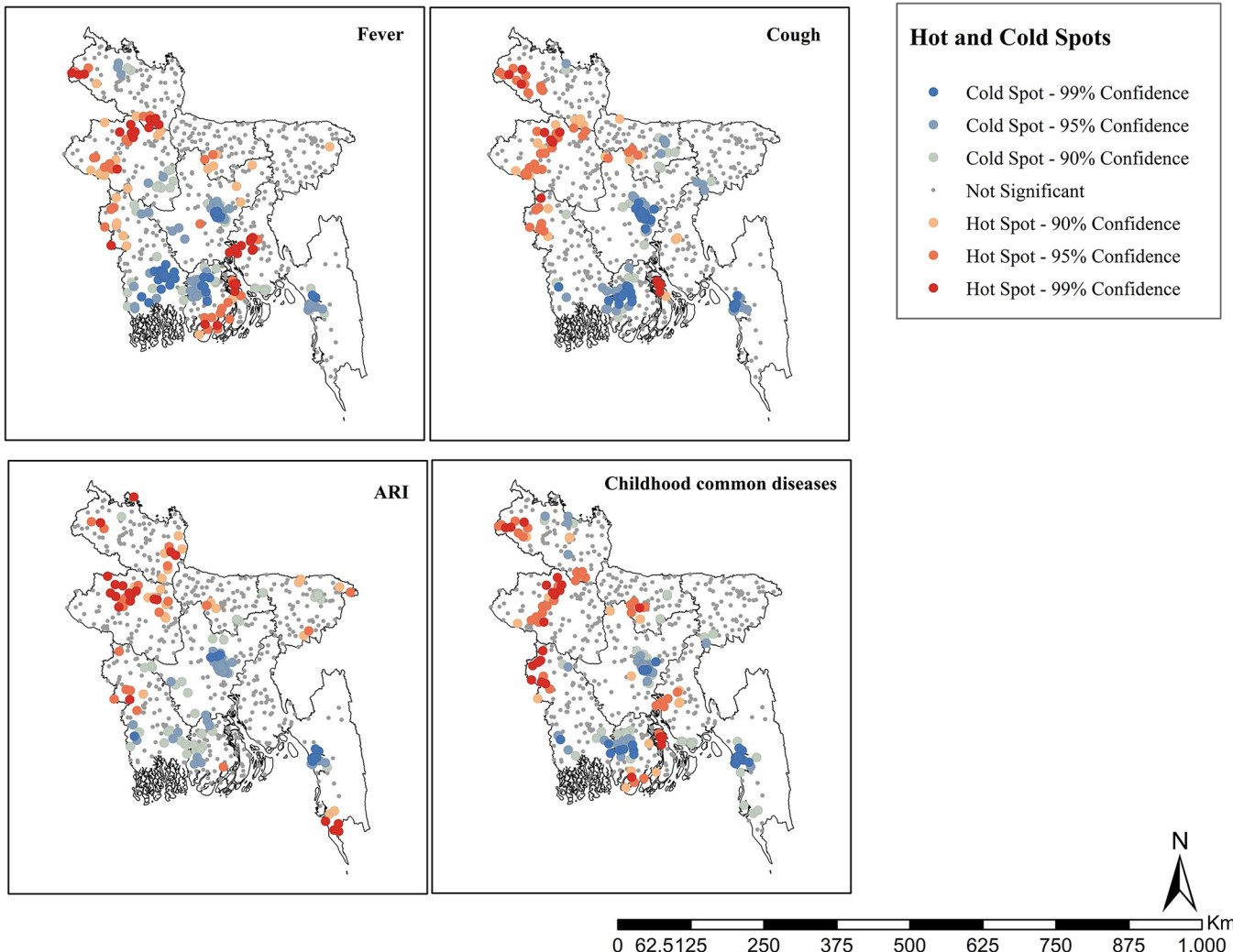

**Fig 4. Hotspot analysis for the prevalence of fever, cough, diarrhea, ARI and childhood common diseases in Bangladesh.** The shape files are freely accessed using the following link: https://data.humdata.org/dataset/cod-ab-bgd.

fever were mainly identified in the Rangpur, Rajshahi, Khulna, Barisal and Chattogram divisions. The prevalence of cough was higher in the north-western, south-western, slightly north-eastern and central-northern parts of the country, and the lower prevalence was in the south-western and south-eastern parts of the country. The estimated higher prevalence of cough in unsampled areas was mainly located in Rangpur, Rajshahi and Khulna divisions and somewhat found in Mymensingh, Barisal and Sylhet divisions (Fig 5).

The estimated areas with the highest prevalence of ARI were found in the north-western part, the south-western part and the very most south-eastern part, while the lower prevalence was remaining in the central, south-western, south-eastern and slightly north-eastern parts. The majority of these estimated areas with a higher prevalence of ARI were observed in the Rangpur, Rajshahi, and Chattogram divisions, as well as slightly in the Khulna division. The estimated prevalence of common childhood illnesses was higher in the central-northern, north-western, south-western and southern parts whereas a lower prevalence was found in the south-eastern and south-western parts. Most of the estimated areas with a higher prevalence of

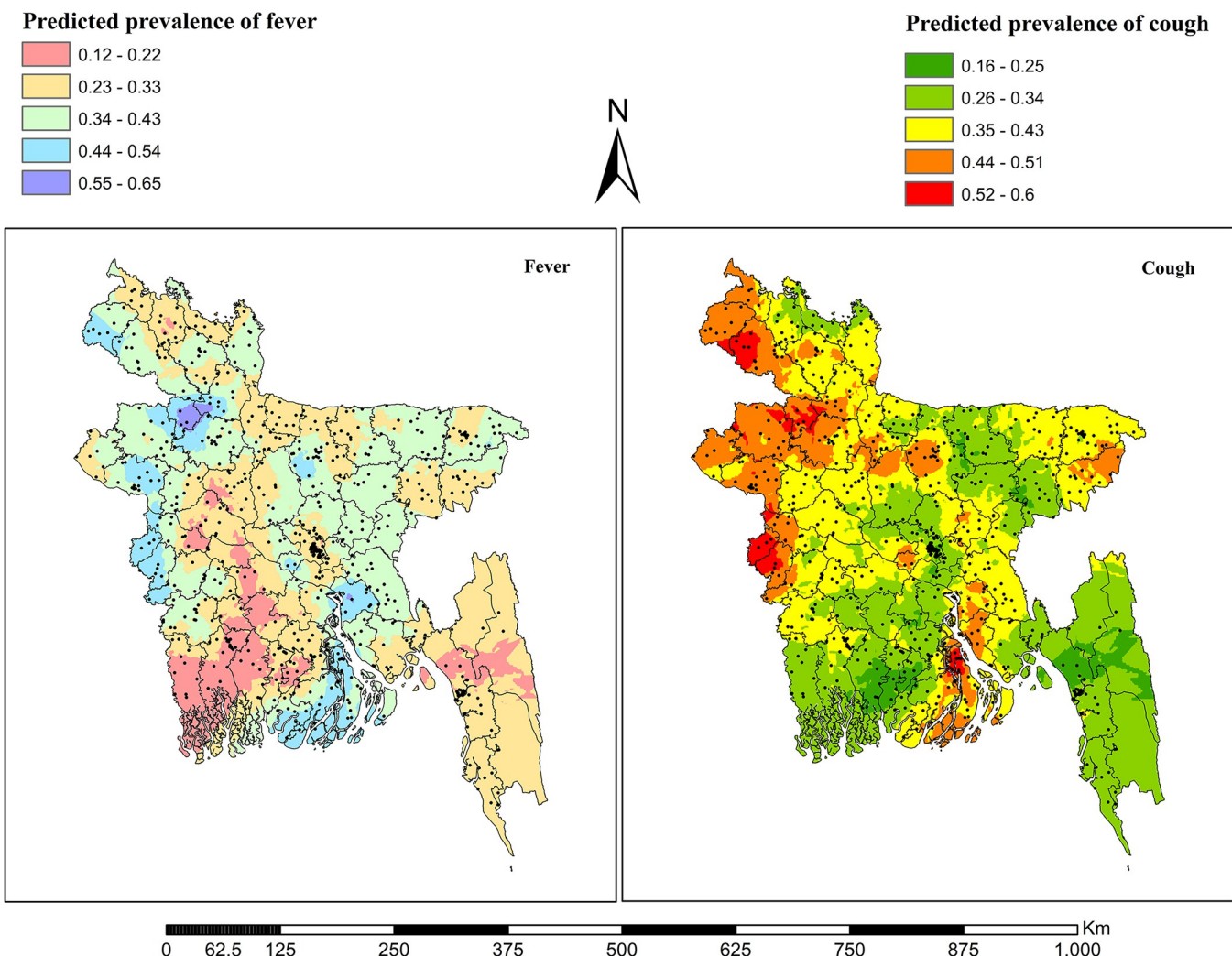

**Fig 5. Kriging interpolation of the spatial clustering of fever and cough in Bangladesh.** The shape files are freely accessed using the following link: https://data.humdata.org/dataset/cod-ab-bgd.

at least one common disease among under five children were observed in the Rangpur, Rajshahi, Khulna, Barisal, Chattogram and Mymensingh divisions (Fig 6).

## Discussion

Utilizing data from BDHS 2017–18, this study aimed to explore the common childhood diseases among children under five years old in Bangladesh in terms of various factors along with spatial distribution. We quantified the common childhood illnesses among children under five years and showed the spatial distribution of these diseases.

The results indicate that as children's age increases, they are less likely to suffer from ARI, a finding that aligns with previous studies conducted in Bangladesh [22,23]. One explanation might be that as children get older, their immunity increases and they learn better ways to deal with their surroundings, such as staying away from dirty places and eating a balanced diet, which suggests that childhood illnesses occur less often as children get older [32,37,38]. The

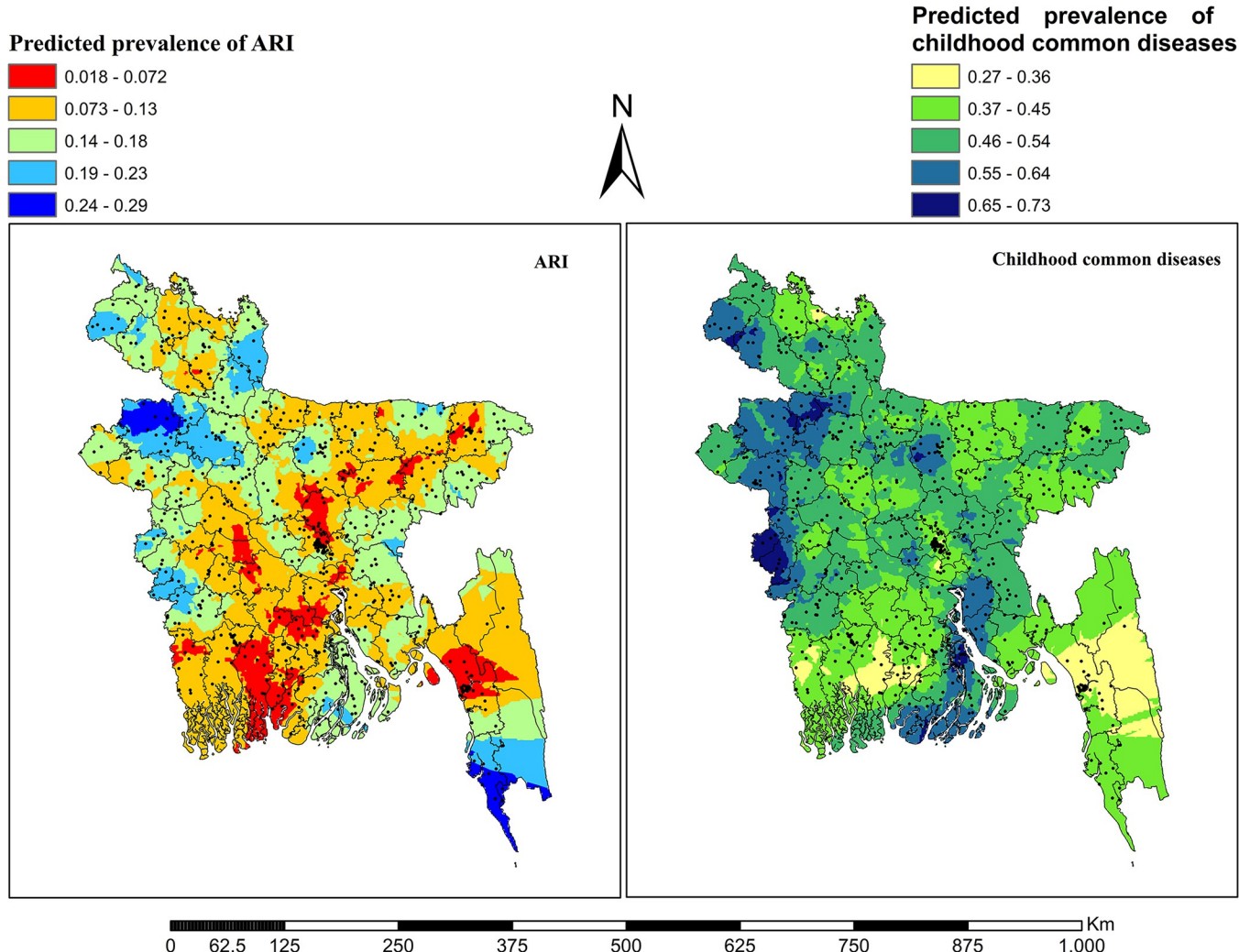

**Fig 6. Kriging interpolation of the spatial clustering of ARI and childhood common diseases in Bangladesh.** The shape files are freely accessed using the following link: https://data.humdata.org/dataset/cod-ab-bgd.

conclusion is that older children should receive more immunizations to prevent common diseases [39].

Our findings suggest that gender may significantly contribute to the onset of certain diseases. Male children were identified as having a higher likelihood of being exposed to ARI than their counterparts, which is supported by several studies conducted in countries on the Asian subcontinent [40–43]. Similar to ARI, a study conducted in Tamil Nadu, India [40] found that the male population was larger and exhibited a higher likelihood of coughing than the female population. We also observe a similar trend in common childhood diseases: female children are less likely than male children to have any common childhood diseases. This could be due to the fact that male children tend to play outside more frequently, which exposes them to infectious aerosols from the surrounding outdoor environment.

After operationalizing our study, we discovered that children from non-Muslim families are less likely to be affected by fever than children from Muslim families, consistent with a study conducted in Bangladesh [23]. We have observed a similar pattern of diarrhea, as well as at least one common childhood disease. This study reveals that responding children from

Muslim families are more prone to suffering from diarrhea than children from non-Muslim families. Research done in India indicated that under-five-year-old children from Muslim homes are 18% more likely to get diarrhea [44], which is significantly lower than the results of this study, which found that this risk is 78% greater among children from Muslim families. This gap in becoming affected by diarrhea or any sickness typically observed in children may develop due to varied beliefs on cleanliness and dietary habits across various religious groups. In order to adhere to the principles of Halal-Haram, the Muslim community follows stringent dietary restrictions that limit their use of some nutritious foods [45]. In addition, Muslim consumers have certain requirements for medical care that vary from those of non-Muslim customers. Specifically, there is an increasing need among Muslim customers for pharmaceutical products that adhere to Halal standards [46]. Therefore, the choice of food and medication may be a contributing reason for the higher incidence of illnesses among children in Muslim families compared to those in non-Muslim families.

We were able to unearth that underweight children are more likely to be susceptible to fever, cough and at least one childhood common disease than their counterparts. Although previous research utilizing the same data set as ours yielded identical results regarding the correlation between underweight status and diarrhea, or the coexistence of fever and diarrhea [44], our study was able to establish a relationship between underweight status and cough, as well as underweight status and at least one type of common childhood illness. Children who have fever and cough symptoms are more likely to become underweight, according to studies from east Africa [47] and Pakistan [48]. Our study supports this conclusion in a different manner. We see a growing correlation between being underweight and having a fever, cough, or at least one of the four prevalent illnesses. Children who are underweight are at increased risk for sickness because their compromised immune systems are unable to effectively fight off pathogens. This is generally the consequence of poor nutrition and overall health.

Children who received the BCG vaccine were more likely to experience fever, congestion, diarrhea, and ARI, based on the findings of this study. This aligns with a previous study conducted in Bangladesh [23]. However, this contradicts the findings of a Ugandan study [49], which found that BCG-vaccinated infants had a diminished risk of duple fever. In addition, the results of this study suggest that children who received the BCG vaccine were more likely to suffer from at least one common childhood illness. Regardless of how undesirable it is. A transient and moderate inflammatory response may be triggered by the BCG vaccine, which may increase the risk of fever, cough, diarrhea and ARI in vaccinated children compared to unvaccinated children.

Our research indicates that there is a correlation between the age of mothers and the susceptibility of children to diarrhea. This finding aligns with a similar study done in Nigeria [50]. One plausible hypothesis is that older mothers show more concern for their children's welfare and take on a more proactive role in protecting them from diseases.

To our astonishment, our study revealed that children whose mothers were educated at secondary school suffered more from fever than children whose mothers were not educated at all. Research in Nigeria [51] indicated that the negative impact of maternal education was substantial in coping with fever in children younger than five. Other studies in Bangladesh [52] and Ethiopia [53] did not find a significant correlation between mothers' levels of education and their children's risk of developing fever. Our study also indicates that children whose moms have completed primary and secondary school are more susceptible to prevalent childhood diseases. While there is anticipation that an increase in maternal education would lead to a decrease in the likelihood of fever and common disease symptoms in children, it is important to acknowledge that this assumption may not always hold true. This discrepancy could be

attributed to a potential disparity between knowledge and actual implementation, particularly in relation to health outcomes [51].

There is a correlation between the mother's age at her first delivery and her children's vulnerability to ARI. Children whose mothers conceived their first child at or before the age of 19 had a higher likelihood of experiencing ARI in comparison to children whose mothers were older than 19 at the time of their first childbirth. The findings of our investigation are corroborated by a study done in Pakistan [54]. Infants delivered to adolescent mothers (aged 10–19) face an elevated risk of being born prematurely, too small, and developing severe health complications upon birth, according to the World Health Organization (WHO) [55]. Therefore, we can hypothesize that offspring born to younger mothers are more likely to have compromised immune systems, which increases their susceptibility to infections like ARI and other ailments.

The results of this research have shown a positive link between mothers' working status and ARI in children under five years old. The findings are consistent with those reported in two other studies: one from Pakistan [56]and the other from Ethiopia [57]. According to the previously cited research, children whose mothers work have a far greater risk of developing ARI than children whose mothers do not work. A plausible explanation can be linked to childcare availability, whereby women play a critical role in children's ARI [58]. Working moms may frequently encounter certain toxins, pollutants, or dangerous vapors during their employment, potentially increasing the risk of infection transfer to their offspring. The shortened nursing window that working mothers experience increases newborns' susceptibility to acute respiratory distress.

Children living in families consisting of one to four individuals had a notably elevated susceptibility to developing coughs in comparison to their counterparts residing in households comprising five to eight people. Perhaps there is a rationale for the heightened level of concern shown by family members for the well-being of children within larger familial units. This may be because more people need care.

An important discovery from our research indicates that children living in households with an average wealth index have a lower level of exposure to ARI compared to the children in our sample who reside in families with the lowest wealth index. On the other hand, one could argue that children from socioeconomically disadvantaged households are more susceptible to ARI than their counterparts from households with moderate income levels. This finding aligns with the outcomes reported in other studies [4,13,14,37]. The lack of financial resources among children from disadvantaged socioeconomic backgrounds results in their inability to get the necessary things, like hygienic food, crucial vaccinations, etc., that are critical for supporting their nutritional well-being. Furthermore, children from low-income families frequently experience unsanitary living conditions, increasing their susceptibility to ARI.

Based on our findings, it was observed that children under the age of five whose families consume water from tube wells are more susceptible to experiencing symptoms such as cough, diarrhea, ARI, and at least one common childhood illness compared to those families who use water from sources other than tube wells. Research conducted in Ghana [59] and Bangladesh [13] has shown that children living in families with better drinking water sources are at a lower risk of exposure to ARI. According to studies conducted in India [60] and Ethiopia [61], there is a higher likelihood of diarrhea and other childhood diseases occurring among children under the age of five whose families have selected tube wells as their primary source of drinking water compared to their counterparts. All the examined studies regarded tube wells as an improved source of drinking water, a view that directly contradicts our research results. One potentially dubious factor is the inadequate testing of the majority of tube wells for water contaminants, which calls into question whether or not they qualify as safe or improved sources of potable water.

Both empirical Bayesian Kriging and ordinary Kriging are effective interpolation techniques that maximize the weight [62]. The predicted prevalence of these illnesses varied by region, according to the predicted percentage of children in the un-sampled locations (enumeration areas) with fever, cough, ARI, and at least one common disease. The north-western and south-western regions of Bangladesh, which mostly include various sections of the Rangpur, Rajshahi, and Khulna divisions, were found to have an estimated higher frequency of fever, cough, ARI, and at least one common illness in unsampled areas. Aside from these, the central-northern portion (in Mymensingh division) had the highest incidence of cough; the lower southern part (in Barisal and Chattogram divisions) had the highest prevalence of fever; and the south-eastern section (in Chattogram division) had the highest frequency of ARI. It is projected that children under five in Bangladesh's central-northern region (in Mymensingh division) and southern region (in Barisal and Chattogram divisions) have higher rates of at least one common childhood illness.

## Limitations and strength

The findings of this study are based on Bangladesh's largest and most recent nationally representative surveys. This approach is applicable to the entire population. Despite its core strengths, this approach has certain limitations that warrant attention. For instance, children under the age of five may experience multiple illnesses, such as ARI and diarrhea, which can be linked to fever or diarrhea and ARI, respectively. This means that one type of disease may occur in the presence of another, potentially having a mixed impact on the respondent. However, in this study, we did not consider the coexistence of specific illnesses due to a lack of time and resources. Furthermore, there may be spatial and temporal variation in the prevalence of the diseases considered in this study.

## Recommendations and policy implications

While the partnership between the government and organizations like SMC and BRAC has reduced diarrhea deaths by increasing ORS availability, ongoing efforts are crucial. To further reduce childhood illness, we must improve intervention coverage and promote nutrition and WASH strategies. In future study, it is recommended that researchers examine the co-occurrence of four prevalent morbidities, as well as the potential influence of confounding variables. Further investigation may be conducted to identify the correlation between various diseases and their probable confounders, while considering different age groups [41].

## Conclusion

Common childhood illnesses are still a concerning matter, with a high prevalence among under-five children in Bangladesh. There are numerous potential factors that are associated with childhood common illnesses, including the child's age in months, the BCG vaccination, the source of potable water, underweight status, the mother's educational level, and the early age of the first birth. This study emphasizes the need to raise parents' educational attainment and improve access to clean water sources in order to improve the nutritional condition of under-five children in Bangladesh.

## Acknowledgments

The authors express their gratitude to the Demographic and Health Surveys (DHS) Program for providing BDHS data access for the research. A special appreciation and sincere gratitude

go to Sharif Rayhan Nafi (web developer) for his unwavering assistance in revising the graphs used in this study.

## Author Contributions

**Conceptualization:** Khondokar Naymul Islam, Sumaya Sultana, Ferdous Rahman, Abdur Rahman.

**Data curation:** Khondokar Naymul Islam.

**Formal analysis:** Khondokar Naymul Islam.

**Methodology:** Khondokar Naymul Islam.

**Supervision:** Abdur Rahman.

**Validation:** Abdur Rahman.

**Writing – original draft:** Khondokar Naymul Islam, Sumaya Sultana, Ferdous Rahman.

**Writing – review & editing:** Abdur Rahman.

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
