## [Decision Letter · Decision Letter 0]

11 Jun 2024

PONE-D-23-39825Exploring the impact of child underweight status on common childhood illnesses among children under five years in Bangladesh along with spatial analysisPLOS ONE

Dear Dr. Rahman,

Thank you for submitting your manuscript to PLOS ONE. After careful consideration, we feel that it has merit but does not fully meet PLOS ONE’s publication criteria as it currently stands. Therefore, we invite you to submit a revised version of the manuscript that addresses the points raised during the review process.

It is recommended to add the crude odds ratio along with AOR in Table 4.

We look forward to receiving your revised manuscript.

Kind regards,

Md. Moyazzem Hossain

Academic Editor

PLOS ONE

3. We note that Figurs 2, 4, 5 and 6  in your submission contain [map/satellite] images which may be copyrighted. All PLOS content is published under the Creative Commons Attribution License (CC BY 4.0), which means that the manuscript, images, and Supporting Information files will be freely available online, and any third party is permitted to access, download, copy, distribute, and use these materials in any way, even commercially, with proper attribution. For these reasons, we cannot publish previously copyrighted maps or satellite images created using proprietary data, such as Google software (Google Maps, Street View, and Earth). For more information, see our copyright guidelines: http://journals.plos.org/plosone/s/licenses-and-copyright.

a. You may seek permission from the original copyright holder of Figurs 2, 4, 5 and 6 to publish the content specifically under the CC BY 4.0 license.  

Reviewers' comments:

Reviewer's Responses to Questions

**Comments to the Author**

1. Is the manuscript technically sound, and do the data support the conclusions?

Reviewer #1: Partly

Reviewer #2: Yes

2. Has the statistical analysis been performed appropriately and rigorously? 

Reviewer #1: No

Reviewer #2: Yes

3. Have the authors made all data underlying the findings in their manuscript fully available?

Reviewer #1: Yes

Reviewer #2: Yes

4. Is the manuscript presented in an intelligible fashion and written in standard English?

Reviewer #1: No

Reviewer #2: Yes

5. Review Comments to the Author

Reviewer #1: Comments to authors

This article offers insights into childhood illness, exploring its prevalence and determinants through spatial analysis. While the merits of the study lie in its spatial analysis, some of the other findings presented are rather conventional and have been previously uncovered by other researchers. The English language in the paper requires significant improvement, as there is a noticeable inconsistency in the use of past and present tenses throughout the text. I have some specific concerns:

1. Abstract is missing important findings regarding spatial analysis. Further, what types of children’s, mother’s and household characteristics are associated with the childhood illness should briefly mentioned.

2. The paragraphs in the introduction are lack of coherence. Very scattered presentation of information not well synchronized.

3. Introduction is lacking intellectuality in organizing the rational of the study. Some studies addressed the relationship between child underweight and childhood illness in Bangladesh. Please see “Islam, M.S.; Chowdhury, M.R.K.; Bornee, F.A.; Chowdhury, H.A.; Billah, B.; Kader, M.; Rashid, M. Prevalence and Determinants of Diarrhea, Fever, and Coexistence of Diarrhea and Fever in Children Under-Five in Bangladesh. Children 2023, 10, 1829.” Authors should add strong justification to counter the objectives.

4. Treatment of missing values employed in this study was not a standard approach.

5. The outcome measures were poorly defined. Did all outcomes clinically diagnose or any treatment approach use to confirm having illness?

6. Authors could use random effect model to determine the childhood illness as the survey followed multistage sampling technique.

7. Most of the findings and interpretations in results and discussion section are well known and previously identified.

8. Strength and limitation of the study is a bit strange. “one is impacted by others, however, in this study we do not consider the mixed illness……………………………. The authors the prevalence of the diseases considered in this study.” Why authors did not do these in this study? The narration of this section was very poorly written and should check grammatical expression.

9. Further the study lacks important recommendations and policy implications.

Reviewer #2: The manuscript is well descriptive with the support of detailed statis analysis. It also include literature from LMICs. However, it is suggestive to add following points:

1. Add "Inclusion Criteria" heading to further understand which populations were added to the study.

2. Add methodology process as to how these populations were contacted and obtained data.

3. Explain why Muslim communities has higher prevelences of fever as compare to non Muslim communities.

4. Add relation/influence of financial status of these families on their buying capacity of hygienic food, impact on nutritional status, vaccination affordability etc

5. Add "Recommendation" heading to expalin strategics/interventions to decrease these high prevelences, role of SDG, NGOs, local health centers, role of government etc.

6. As mentioned in the limitation, further research is required to explore relation of BCG and fever, it is highly advisable to researchers to conduct such study. This is unfold many causes such as and might be problem to cold chain maintainance, manufacturong of BCG vaccine, technique of administration and comparing such data with other LMICs.

6. PLOS authors have the option to publish the peer review history of their article (what does this mean?). If published, this will include your full peer review and any attached files.

Reviewer #1: No

Reviewer #2: **Yes: **Zaibunissa karim

---

## [Author Response · Author response to Decision Letter 0]

9 Aug 2024

Reviewer’s comment 

Reviewer #1: Comments to authors

This article offers insights into childhood illness, exploring its prevalence and determinants through spatial analysis. While the merits of the study lie in its spatial analysis, some of the other findings presented are rather conventional and have been previously uncovered by other researchers. The English language in the paper requires significant improvement, as there is a noticeable inconsistency in the use of past and present tenses throughout the text. I have some specific concerns:

1. Abstract is missing important findings regarding spatial analysis. Further, what types of children’s, mother’s and household characteristics are associated with the childhood illness should briefly mentioned. 

Reply: The findings regarding spatial analysis are now added to the abstract. Additionally, children’s, mothers’ and household characteristics that are associated with the childhood illnesses are also mentioned briefly in the abstract, as per your suggestion.

2. The paragraphs in the introduction are lack of coherence. Very scattered presentation of information not well synchronized.

Reply: We have tried to reorganize the writing style of the paragraphs in the introduction section so it ensures smooth flow and the synchronization.

3. Introduction is lacking intellectuality in organizing the rational of the study. Some studies addressed the relationship between child underweight and childhood illness in Bangladesh. Please see “Islam, M.S.; Chowdhury, M.R.K.; Bornee, F.A.; Chowdhury, H.A.; Billah, B.; Kader, M.; Rashid, M. Prevalence and Determinants of Diarrhea, Fever, and Coexistence of Diarrhea and Fever in Children Under-Five in Bangladesh. Children 2023, 10, 1829.” Authors should add strong justification to counter the objectives.

Reply: We have made necessary corrections to the introduction, rewrote the necessary subsections to enhance the study's intellectual depth and ensure its rationality. Though some studies addressed the relationship between child underweight and childhood illnesses, they did not consider all four illnesses as we do. The study you suggested us to review only deals witha fever, diarrhea, and the co-occurrence of these two conditions. However, we have additionally identified the association between child underweight status and cough and the existence of at least one of the four common illnesses among under-five children in Bangladesh, which is a new finding to address.

4. Treatment of missing values employed in this study was not a standard approach.

Reply: Since our main focus was on identifying the geographical distribution of childhood common diseases, we excluded the missing values of the study variables (fever, cough, diarrhea, and ARI) and the missing values of the explanatory variable, which were less than 1% of the dataset. We merged the missing values from more than 1% of the dataset and the "don't know" categories into a new category caled "#Others," this ensures the prevalence of the illnesses to appear more appropriate. In order to create the "#Others" category, we reviewed several other existing studies, such as 

"Kundu S, Kundu S, Seidu AA, Okyere J, Ghosh S, Hossain A, Alshahrani NZ, Banna MH, Rahman MA, Ahinkorah BO. Factors influencing and changes in childhood vaccination coverage over time in Bangladesh: a multilevel mixed-effects analysis."

5. The outcome measures were poorly defined. Did all outcomes clinically diagnose or any treatment approach use to confirm having illness?

Reply: All of the outcomes were self-reported by the mothers in the BDHS dataset, confirming children's diseases according to BDHS reports (https://dhsprogram.com/publications/publication-FR344-DHS-Final-Reports.cfm). To address this issues we have mentioned it under the "outcome variables" section.

6. Authors could use random effect model to determine the childhood illness as the survey followed multistage sampling technique.

Reply: Since the main focus of our study was to find the spatial pattern of the common childhood diseases, we did not perform a random effect model. However, we focused on estimating the prevalence of childhood common diseases in unsampled areas.

7. Most of the findings and interpretations in results and discussion section are well known and previously identified.

Reply: Most of the findings and interpretations in the results and discussion section are well known and previously identified based on the applied logistic regression model's odds ratios. Since no other Bangladeshi study worked with the geographical distribution of illnesses which we were focusing. Additionally, in line with our goal, our study reveals a unique relationship between children's underweight status and cough, as well as between children's underweight status and the presence of at least one of four common illnesses.

8. Strength and limitation of the study is a bit strange. “One is impacted by others, however, in this study we do not consider the mixed illness……………………………. The authors the prevalence of the diseases considered in this study.” Why authors did not do these in this study? The narration of this section was very poorly written and should check grammatical expression.

Reply: Studying the coexistence of different illnesses is crucial for gaining a deeper understanding of the factors that associate with childhood illnesses. Due to a lack of time and resources, we did not consider different combinations of other illnesses as our study variables. However, we tried to enhance both the writing style and grammatical expression in the section's narration.

9. Further the study lacks important recommendations and policy implications.

Reply: New recommendations and potentially important policy implications are added in the revised manuscript.

Reviewer #2: The manuscript is well descriptive with the support of detailed statistics analysis. It also include literature from LMICs. However, it is suggestive to add following points:

1. Add "Inclusion Criteria" heading to further understand which populations were added to the study.

Reply: According to your suggestions, a new section is added in the revised manuscript under the heading called "Inclusion Criteria".

2. Add methodology process as to how these populations were contacted and obtained data.

Reply: As per your suggestion, we have added the process of sampling and contacting the population to obtain data to the methodology section.

3. Explain why Muslim communities has higher prevalence of fever as compare to non-Muslim communities.

Reply: Based on previous scientific studies, a tentative explanation for the higher prevalence of fever and other illnesses among Muslim children compared to non-Muslim children is established, which we also addressed in the revised manuscript.

4. Add relation/influence of financial status of these families on their buying capacity of hygienic food, impact on nutritional status, vaccination affordability etc.

Reply: Following your suggestion, we have mentioned the influence of these families' financial status on their ability to purchase hygienic food, its impact on their nutritional status, and the affordability of vaccinations in the revised version of the manuscript.

5. Add "Recommendation" heading to explain strategies/interventions to decrease these high prevalence, role of SDG, NGOs, local health centers, role of government etc.

Reply: We have included the section titled "Recommendation," which suggests involving SDGs, NGOs, local health clinics, and the government as a means to decrease the widespread occurrence of common infant illnesses.

6. As mentioned in the limitation, further research is required to explore relation of BCG and fever, it is highly advisable to researchers to conduct such study. This is unfolding many causes such as and might be problem to cold chain maintenance, manufacturing of BCG vaccine, technique of administration and comparing such data with other LMICs.

Reply: We regret the inconvenience caused by the limitations of this study; additional research is required to investigate the correlation between BCG and prevalent childhood illnesses. In this study, we conducted a thorough examination of the BCG vaccination's status in order to ascertain its influence on the aforementioned common childhood illnesses. We have removed the statement from the section and have modified it accordingly.

---

## [Decision Letter · Decision Letter 1]

16 Sep 2024

Exploring the impact of child underweight status on common childhood illnesses among children under five years in Bangladesh along with spatial analysis

PONE-D-23-39825R1

Dear Dr. Rahman,

We’re pleased to inform you that your manuscript has been judged scientifically suitable for publication and will be formally accepted for publication once it meets all outstanding technical requirements.

Kind regards,

Md. Moyazzem Hossain

Academic Editor

PLOS ONE

Additional Editor Comments (optional):

Reviewers' comments:

Reviewer's Responses to Questions

**Comments to the Author**

1. If the authors have adequately addressed your comments raised in a previous round of review and you feel that this manuscript is now acceptable for publication, you may indicate that here to bypass the “Comments to the Author” section, enter your conflict of interest statement in the “Confidential to Editor” section, and submit your "Accept" recommendation.

Reviewer #1: All comments have been addressed

2. Is the manuscript technically sound, and do the data support the conclusions?

Reviewer #1: Partly

3. Has the statistical analysis been performed appropriately and rigorously? 

Reviewer #1: Yes

4. Have the authors made all data underlying the findings in their manuscript fully available?

Reviewer #1: Yes

5. Is the manuscript presented in an intelligible fashion and written in standard English?

Reviewer #1: No

6. Review Comments to the Author

Reviewer #1: The authors have addressed all comments. I do not have further comments to authors. Now the decision is up to editor for further action.

7. PLOS authors have the option to publish the peer review history of their article (what does this mean?). If published, this will include your full peer review and any attached files.

Reviewer #1: No

---

## [Editor Report · Acceptance letter]

18 Sep 2024

PONE-D-23-39825R1 

PLOS ONE

Dear Dr. Rahman, 

I'm pleased to inform you that your manuscript has been deemed suitable for publication in PLOS ONE. Congratulations! Your manuscript is now being handed over to our production team.

Kind regards, 

on behalf of

Professor Md. Moyazzem Hossain 

Academic Editor

PLOS ONE